# Multimodal Lego: Model Merging and Fusion Across Topologies and Modalities in Biomedicine

**Konstantin Hemker**[1], **Nikola Simidjievski**[2,1] **& Mateja Jamnik**[1]
[1]Department of Computer Science & Technology
[2]PBCI, Department of Oncology
University of Cambridge
Cambridge, UK
`{kh701, ns779, mj201}@cam.ac.uk`

## Abstract

Learning holistic computational representations in physical, chemical or biological systems requires the ability to process information from different distributions and modalities within the same model. Thus, the demand for multimodal machine learning models has sharply risen for modalities that go beyond vision and language, such as sequences, graphs, time series, or tabular data. While there are many available multimodal fusion and alignment approaches, most of them require end-to-end training, scale quadratically with the number of modalities, cannot handle cases of high modality imbalance in the training set, or are highly topology-specific, making them too restrictive for many biomedical learning tasks. This paper presents Multimodal Lego (MM-Lego), a general-purpose fusion framework to turn any set of encoders into a competitive multimodal model with no or minimal fine-tuning. We achieve this by introducing a wrapper for any unimodal encoder that enforces shape consistency between modality representations. It harmonises these representations by learning features in the frequency domain to enable model merging with little signal interference. We show that MM-Lego 1) can be used as a model merging method which achieves competitive performance with end-to-end fusion models without any fine-tuning of the original encoder weights, 2) can operate on any unimodal encoder, and 3) is a model fusion method that, with minimal fine-tuning, surpasses all benchmarks in five out of seven datasets.

## 1 Introduction

The utility and demand for multimodal machine learning approaches has sharply risen due to their potential to derive holistic representations in various systems, including physics (Zubatiuk & Isayev, 2021), chemistry (Belianinov et al., 2018), neuroscience (Alberdi et al., 2016), or biology (Boehm et al., 2022). Multimodal models in the vision & language domains leverage the same data distributions, which are represented across different modalities (Yu et al., 2019; Li et al., 2021b; Tu et al., 2023), such as vision-text pairs of the same concepts. However, in many biomedical domains, modalities represent data at different scales (e.g., cellular, genomic, transcriptomic), cardinalities that are not paired (e.g., many single-cell reads for a single tissue slide per patient), and follow separate distributions. While large foundation models have excelled in tasks confined to individual modalities (Cui et al., 2024; Nguyen et al., 2024; Chen et al., 2024), training these models across modalities is expensive, requires paired modalities, and is still an end-to-end process.

Multimodal *fusion* methods (Zadeh et al., 2017; Liu et al., 2018; Nagrani et al., 2021; Li et al., 2021a) attempt to derive a common representation from different data structures and distributions whilst preserving its salient signals (Baltrusaitis et al., 2019). However, there are several shortcomings of existing fusion methods that relate to their utility, scalability and underlying data assumptions. First, many fusion methods require end-to-end training of the multimodal model, and even in scenarios with existing unimodal models, the fusion operation still has to be trained for a downstream task,

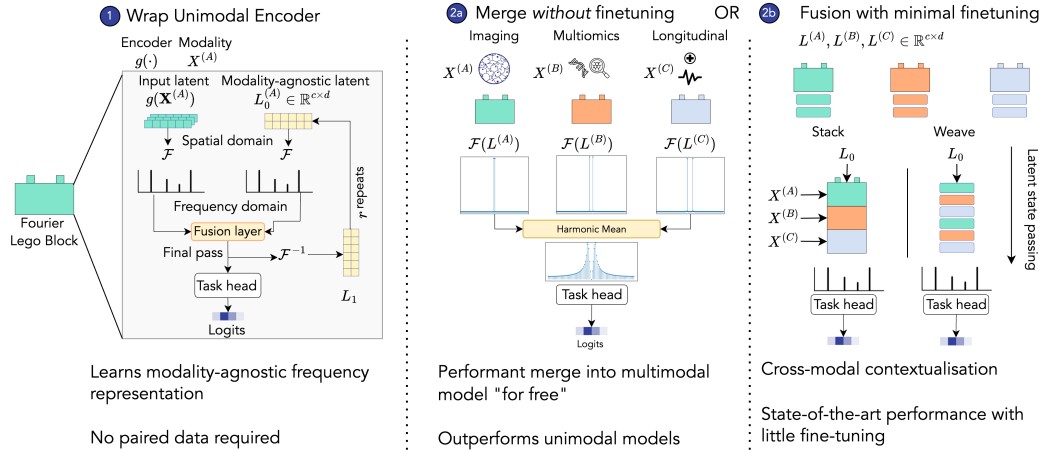

Figure 1: The Multimodal Lego workflow to turn a set of encoders into a performant multimodal model. *LegoBlock* (1) makes unimodal encoders compatible with model merging techniques by learning a latent representation in the frequency-domain to prevent signal interference effects upon aggregation. Any set of *LegoBlocks* can be merged into a multimodal model without any fine-tuning (*LegoMerge* (2a)) or with minimal fine-tuning to achieve state-of-the-art performance (*LegoFuse* (2b)).

typically in a supervised manner. This leads to redundant computational overhead and an inability to extend the model with additional modalities after it has been trained. Second, many commonly used fusion methods either scale quadratically (with the number of modalities) (Zadeh et al., 2017; Chen et al., 2022) or are only designed to operate with two modalities (Chen et al., 2021; Xu & Chen, 2023). Third, many of these methods follow a monolithic design, requiring all modalities to be available for every sample during training. This means that they are not robust to missing modalities, modality imbalance, or non-overlapping training sets, which is a very common challenge in a variety of real-world settings (Swamy et al., 2023). Finally, many end-to-end fusion architectures are highly topology-specific, making them difficult to extend to other domains.

Some of these challenges can be addressed through *model merging* (Labonne, 2024) (also referred to as *knowledge fusion* (Jin et al., 2023)), an approach commonly used in the context of multi-task settings and language modelling, which capitalises on combining well-performing unimodal models trained in isolation. Model merging methods attempt to combine two architecturally identical models trained on different distributions through interpolation, arithmetic manipulation, and aggregation of their weights (Yadav et al., 2023; Stoica et al., 2024; Ilharco et al., 2023), or stacking their layers (Akiba et al., 2024), often without additional training/fine-tuning. While model merging has been extended to some multimodal vision and language tasks (Sung et al., 2023), its crucial challenges in a multimodal setting are that: a) the merged components are still trained in isolation, and b) we cannot assume topological equivalence between two models for separate modalities due to their separate input shapes.

In this paper, we present Multimodal Lego (*MM-Lego*) – a flexible framework for combining any unimodal models into a multimodal model with no or minimal fine-tuning (Figure 1). We introduce two approaches within our framework – *LegoFuse* and *LegoMerge*, enabling performant multimodal models given a set of unimodal encoders with either little (*LegoFuse*) or no (*LegoMerge*) fine-tuning. We show that MM-Lego satisfies multiple desirable properties in a range of real-world multimodal applications combining imaging, tabular, and time series modalities. We demonstrate the utility of MM-Lego on seven medical datasets across three separate downstream tasks, showing that it is:

1. **Performant without end-to-end training**: *LegoMerge* is highly computationally efficient and achieves competitive performance wrt. state-of-the-art *without any fine-tuning*, while *LegoFuse* exceeds the state-of-the-art in some tasks with only a few epochs of fine-tuning.

2. **Scalable**: Both variants of MM-Lego scale linearly with the number of modalities whilst outperforming methods with quadratic time complexity.

3. **Topology agnostic**: Unlike most model merging approaches, *LegoMerge* does not require equivalent architectures between the merged models, allowing users to take advantage of the plethora of open-source models for multimodal learning.
4. **Handling modality imbalance & non-overlapping sets**: MM-Lego is robust in cases of missing modalities and strong modality imbalance, a common problem in medical domains. MM-Lego can be used even if each modality was trained on unpaired (non-overlapping) training samples.

## 2 BACKGROUND & RELATED WORK

**Preliminaries.** Let $\mathbf{X}^{(\mathcal{M})} = \bigcup_{m \in \mathcal{M}} m$ be a multimodal dataset where $\mathcal{M} = \{A, B, \ldots, Z\}$ represents the set of modalities $m$ such as images ($A$), tabular data ($B$), time series ($C$), etc. Let $\mathbf{X}^{(A)}_{i,j,k}$ correspond to the element in the dataset for modality $A$ at sample $i$, column $j$, and channel $k$, assuming $A \in \mathbb{R}^{I \times J \times K}$ where $1 \leq i \leq N, 1 \leq j \leq J, 1 \leq k \leq K$. Each sample in $\mathbf{X}$ has a set of discriminative task labels $\mathbf{y}^{(\mathcal{T})} = \bigcup_{t \in \mathcal{T}} \mathbf{y}^{(t)}$, where $\mathcal{T} = \{T_1, T_2 \ldots, T_c\}$ is the set of possible tasks such that $\mathbf{y}^{(T_1)} = \{y_1^{T_1}, y_2^{T_1}, ..., y_N^{T_1}\}$ are the scalar target values for task $T_1$ for $N$ samples.

**Fusion methods.** Given multiple data inputs or latent representations, fusion methods construct a single learning representation that can be used for downstream tasks, often whilst reducing dimensionality. Many fusion methods (Alberdi et al., 2016; Chen et al., 2022) first learn a set of modality-specific encoders $\mathcal{G} = \{g_m : m \to \mathbf{h}^{(m)}\}$ assuming a single task label $\mathbf{y}$. This results in a set of latent representations $\mathcal{H} = \{g_m(m), m \in \mathcal{M}\}$, which are combined with a fusion operator to obtain the final fused representation $\mathbf{z} = \psi(\mathcal{H})$ and its final prediction $\hat{\mathbf{y}} = f(\mathbf{z})$. Following this problem setup, fusion methods can be differentiated by: 1) the choice of the *fusion operator* $\psi(\cdot)$; 2) the *fusion stage* in the pipeline of when $\psi(\cdot)$ is applied; and 3) the *fusion order* in which the fusion operations are applied (i.e., sequential vs. parallel). The *fusion operator* can be either static (e.g., concatenation (Jaegle et al., 2021), Kronecker product (Zadeh et al., 2017)) or learnable (e.g., low-rank tensor fusion (Liu et al., 2018), cross-attention mechanisms (Nagrani et al., 2021; Li et al., 2021a; Xu & Chen, 2023), mixture of experts (Mustafa et al., 2022; Han et al., 2024)). The *fusion stage* is typically characterised as early, intermediate or late fusion. Early fusion methods apply a static fusion operator $\psi(\cdot)$ to the raw data while only applying this after passing each modality through $\mathcal{G}$. Intermediate fusion methods often do not apply a static aggregation but rather learn a fusion function (i.e., a small sub-network or neural layer) in the latent space as part of its end-to-end training (Baltrusaitis et al., 2019).

A shortcoming of existing fusion methods for many real-world applications is the *fusion order* – most fusion methods use a monolithic task setup, where all modalities are required during training and inference to calculate the set of latent representations $\mathcal{H}$. This often leads to noisy fused representations in cases when a modality is (partially or fully) missing, as the missing tensor requires imputation. Moreover, $\psi(\{g_m(m, \mathbf{y}), m \in \mathcal{M}\})$ is typically trained end-to-end. This hinders the potential of extending the multimodal model with additional modalities (beyond the ones it has been trained on), without training anew. Rerunning the complete training pipeline just to add one (or more) additional modality can be infeasible or can lead to redundant computational overheads. Finally, many methods are highly domain-specific and are either not designed for $|\mathcal{M}| > 2$ or scale quadratically with the number of modalities (Li et al., 2021a; Chen et al., 2021; Xu & Chen, 2023).

**Model merging.** The core idea behind model merging, typically deployed in multi-task settings, is that earlier layers in a network may learn similar features that may be used across tasks (Sundar et al., 2024). Using linear interpolation (Labonne, 2024) or arithmetic manipulation (Ilharco et al., 2023) of the task-specific weights, model merging approaches have shown that they can effectively generalise to new tasks without any fine-tuning. Formally, given multiple tasks $\mathbf{y}^{(\mathcal{T})}$ for the same modality $A$, they first learn the set of task-specific functions $\mathcal{F} = \{f_t(\omega_t) : A \to \hat{\mathbf{y}}^{(t)} \mid t \in \mathcal{T}\}$ where $\omega$ denotes the corresponding model parameters. Assuming the same architecture for each model in $\mathcal{F}$, parameters from a pre-trained base model $\omega_{base}$ can be used to derive task vectors as $\mathcal{V} = \{\tau_t \leftarrow \omega_t - \omega_{base} \mid t \in \mathcal{T}\}$ (Ilharco et al., 2023) . Given these task vectors, a multi-task model can be constructed by updating the weights of the base model $\omega' = \omega_{base} + \lambda \sum_t^{\mathcal{T}} \tau_t$. This idea has been extended by the TIES (Yadav et al., 2023) and DARE (Yu et al., 2024) that merge models through sparsifying and resolving sign conflicts in the task vectors. Another popular approach is spherical linear interpolation (SLERP), a method used to smoothly interpolate between two vectors while respecting the geometric properties of the vector space (Labonne, 2024). More specifically,

given model parameters $\omega_{T_1}$ and $\omega_{T_2} \in \mathbb{R}^d$, derived from two models with identical architectures, the merged multi-task model parameters can be calculated as $\omega' = \omega_{T_1} \frac{sin(\theta \cdot (1-\mu))}{sin(\theta)} + \omega_{T_2} \frac{sin(\theta \cdot \mu)}{sin(\theta)}$ where $\theta$ is the radian between the two vectors $\omega_{T_1}$ and $\omega_{T_2}$ (Shoemake, 1985). The underlying assumption of the above model merging approaches is that the models should have an equivalent network topology, ensuring that the dimensions $\mathbb{R}^d$ match up between tasks. However, while this is an acceptable constraint for multi-task learning, it is infeasible for multimodal models where modality shapes and the corresponding network topologies vary greatly.

## 3  MULTIMODAL LEGO

MM-Lego introduces two novel approaches for multimodal model merging (*LegoMerge*) and multimodal model fusion (*LegoFuse*). It addresses a number of limitations of existing model merging (topological equivalence) and fusion methods (such as scalability for $|M| > 2$, missing modality handling, end-to-end training, paired data, etc.). The core component of MM-Lego is a *LegoBlock* - a wrapper for any modality-specific encoder that imposes several constraints on the latent feature space and structure. These constraints are a necessary condition for *LegoMerge*, which aggregates the latent encodings with minimal signal interference between modalities to perform a multimodal model merge. Moreover, using *LegoFuse*, MM-Lego allows us to fine-tune the combined blocks, ensuring that cross-modal dependencies and mutual context can be learned.

**Architecture.** Rather than learning a single fusion operator $\psi(\mathcal{H})$ that applies to all latent representations at once, we learn a set of latent update functions for each modality, in the form of

$$\mathcal{B} = \{\psi_m : (g_m(X^{(m)}), L_s^{(m)}) \rightarrow L_{s+1}^{(m)} \mid s \in S, m \in \mathcal{M}\}, \tag{1}$$

where $L_t^{(m)} \in \mathcal{L}$ is our target latent representation for each modality that we will later use in the merge and fusion, and $S$ is the number of update steps. Using the iterative update architecture with latent state passing in Equation 1 has a number of advantages. First, iterative attention architectures have been shown to be highly generalisable across modalities (Jaegle et al., 2022), and effective at dealing with missing individual modalities (Swamy et al., 2023; Hemker et al., 2024). Second, since all modalities are encoded into a self-defined latent representation, we can impose a dimensionality constraint such that each latent has the same dimensions (e.g., $L^{(A)}, L^{(B)}, L^{(C)} \in \mathbf{R}^{c \times d}$ for latent channels and dimensions $c$ and $d$). Third, we can do latent state passing between elements in $\mathcal{B}$, which allows us to "stack" the update functions on top of each other (hence the name) to sequentially encode each modality's signal into the same latent representation.

**LegoBlocks.** Each element in $\mathcal{B}$ represents a *LegoBlock* (Figure 2), which learns the latent update function $\psi_m$ for any given encoder $g_m$. Acknowledging that different data modalities and structures require different inductive biases to effectively encode each modality's information ($g_m$), *LegoBlock* acts as a wrapper to accurately encode $h_m$ into $L^{(m)}$. The benefit of training each modality update function separately instead of end-to-end is that we can train on entirely separate samples for the

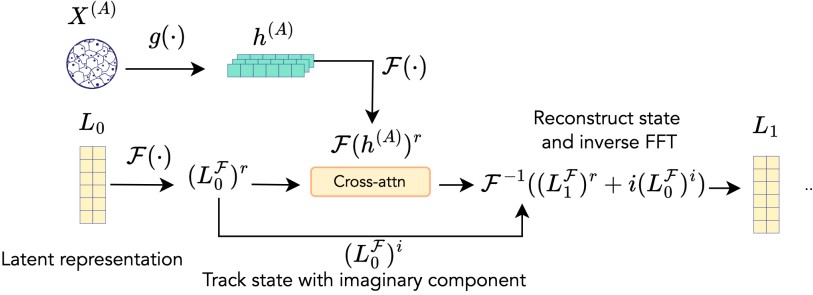

Figure 2: Frequency-domain state passing in *LegoBlock*. The latent bottleneck $L_0$ is randomly initialized as a model parameter at the start of training and iteratively updated by each pass through the *LegoBlocks*. The real components of the FFT $(L_0^{\mathcal{F}})^r$ and $\mathcal{F}(h^{(A)})^r$ are used in the cross-attention update, and the imaginary component $(L_0^{\mathcal{F}})^r$ is used for reconstruction.

same tasks. For example, in many medical domains, we may have single-cell data for one subset of patients and bulk sequencing data for a different subset, while having the same task labels for the entire set. To address this, we use latent representations $L$ that effectively encode signal across modalities, and are robust or invariant to transformations (shifts, rotations, etc.), noise and signal interference. Alleviating signal interference is particularly important for model merging, as it is undesirable to apply an aggregate function on the learned modality latents $\mathcal{L}$ that can cancel out each other's signal. Beyond preventing signal interference, Fourier features have also been shown to be effective as mixing mechanisms (Lee-Thorp et al., 2021), positional encodings (Lee-Thorp et al., 2021), and to be naturally emerging in invariant networks (Marchetti et al., 2023).

This motivated us to design MM-Lego for learning latent representations in the frequency domain, taking advantage of a number of desirable properties for multimodal merging and fusion. In particular, frequency-domain representations are: 1) *signal-preserving* as frequency features are less prone to signal interference upon aggregation (see Appendix F); 2) *distance-preserving*, as the Euclidean distance between two signals remains unchanged after the Fourier Transform (following from Parseval's Theorem (Parseval, 1806)), making them suitable for distance-based loss functions; 3) *invertible* as the spatial/temporal domain can be reconstructed, allowing for the iterative updates outlined in Equation 1; and 4) *efficient*, as the Fast Fourier Transform (FFT) has a time complexity of $O(n\,log(n))$, making it scalable to very large datasets (Lee-Thorp et al., 2021). Further detail on why the Fourier transform exhibits desirable properties for model mergin can be found in Appendix B.

Figure 2 depicts a single update of *LegoBlock* that operates over frequency-domain latent of modality $X^{(A)}$. Starting with the latent representation in the spatial domain, we first apply a discrete FFT $\mathcal{F}(\cdot)$ (Nussbaumer, 1982) along each dimension of the 2D Tensor to yield a frequency domain representation:

$$L_t^{\mathcal{F}}(u, v) = \sum_{i=0}^{c-1}\sum_{j=0}^{d-1} L_t(i,j)e^{-2\pi i(\frac{ux}{c}+\frac{vy}{d})}, \tag{2}$$

where $i, j$ denote the spatial-domain indices, and $u, v$ denote the frequency-domain indices. This results in a complex frequency-domain representation from which we separate the real (symmetrical) and imaginary (asymmetrical) components of the FFT ($(L_t^{\mathcal{F}})^r$ and $(L_t^{\mathcal{F}})^i$) (Smith et al., 1997). We update the real component using a standard cross-attention layer (Vaswani et al., 2017), where we aim to learn the weight matrices $W_m^q$ for the update query $(L_t^{\mathcal{F}})^r$, and $W_m^k$, $W_m^v$ for the keys and values ($h^{(A)}$) resulting in the latent update:

$$(L_{t+1}^{\mathcal{F}})^r = \text{softmax}\left(\frac{(L_t^{\mathcal{F}})^r W_m^q \cdot (\mathcal{F}(h^{(A)})^r W_m^k)^\top}{\sqrt{d_k}}\right) \cdot (\mathcal{F}(h^{(A)})^r W_m^v). \tag{3}$$

In contrast to other Fourier-based architectures (Lee-Thorp et al., 2021), which only use the real component of the transform, we keep track of the imaginary component $(L_t^{\mathcal{F}})^i$ as well. This allows us to reconstruct the complex representation, and subsequently apply the inverse transform. We found this to be critical for our iterative architecture, as otherwise the signal gets distorted and we lose phase information (encoded in the imaginary component) at each update pass. Once we reconstruct the complex representation, we apply the inverse transform to recover the spatial representation in preparation of the next pass $L_{t+1} = \mathcal{F}^{-1}((L_{t+1}^{\mathcal{F}})^r + i(L_t^{\mathcal{F}})^i)$. Finally, the last task-specific heads of each block are a fully-connected layer after applying layer normalisation. We omit the inverse transform after the last update such that each head is trained in the frequency domain. This ensures that we can apply aggregations with low signal interference on $\mathcal{L}$ during *LegoMerge*.

**LegoMerge.** With the architectural assumptions imposed on each modality encoder in $\mathcal{G}$ through *LegoBlocks* $\mathcal{B}$, we can apply model merging techniques in a multimodal setting. With $\mathcal{L} \subseteq \mathbb{R}^{c \times d}$ and each element in $\mathcal{L}$ being in the frequency domain, we can use aggregation functions $\psi(\cdot)$, which are less prone to cancelling out signal. For example, let $L^{(A)}$ and $L^{(B)}$ be the final frequency domain latent representations for modalities $A$ and $B$, then we can calculate a merged multimodal representation as:

$$\psi(L^{(A)}, L^{(B)}) = (\frac{2|L^{(A)}| \cdot |L^{(B)}|}{|L^{(B)}| + |L^{(A)}|}) \cdot e^{i \cdot \frac{\angle L^{(A)} + \angle L^{(B)}}{2}}, \tag{4}$$

where the real component is the harmonic mean of the magnitudes ($|\cdot|$), and the imaginary component is the arithmetic mean of the phases ($\angle$) of $L^{(A)}$ and $L^{(B)}$. We take the harmonic mean since it is less prone to outliers (Smith et al., 1997), that is, the merged representation is less likely to be strongly

skewed towards either modality by very large frequency components. With the cross-modal combined representation $L^{(\mathcal{M})}$, we need to combine the task heads of each block, where we apply spherical linear interpolation (SLERP) (Shoemake, 1985) for the set of task heads $\mathcal{Y}$ from each element in $\mathcal{B}$. Assuming two task heads with weight vectors $\mathbf{w}_{y1}$ and $\mathbf{w}_{y2}$, where $\mathbf{w}_{y1} \subset \omega_A, \mathbf{w}_{y2} \subset \omega_B$, we calculate the merged weights as $\mathbf{w}_{y1y2} = \mathbf{w}_{y1} \cdot \frac{sin(\theta \cdot (1-\mu))}{sin(\theta)} + \mathbf{w}_{y2} \cdot \frac{sin(\theta \cdot \mu)}{sin(\theta)}$ (Shoemake, 1985), where $\theta$ is the angle (in radians) between both vectors and $\mu$ is a binary variable indicating whether $\mathbf{w}_{y1}$ or $\mathbf{w}_{y2}$ is used.

**LegoFuse.** *LegoMerge* is designed to construct a performant multimodal model without any additional training. Nevertheless, its key limitation is that each element in $\mathcal{B}$ is trained in isolation. To allow for modalities to mutually contextualise each other, a limited amount of fine-tuning is beneficial. To avoid fine-tuning a potentially noised signal emerging from the merged latent $L^{(\mathcal{M})}$, rather than directly fine-tuning the merged model (at the parameter-level), *LegoFuse* operates at the layer level by sequentially passing through all layers in $\mathcal{B}$. Specifically, the shape consistency introduced by $\mathcal{L} \subseteq \mathbb{R}^{c \times d}$ allows the stacked model to pass the Fourier-transformed latent states either between blocks (stacking) or different layers between blocks (weaving), as illustrated in Figure 1. We then fine-tune the stacked or the weaved model for a few epochs with all (paired) modalities, such that the state updates are conditioned on all modalities' updates. This, in turn, becomes the query for the cross-attention layer. Note that both the stacked and weaved variants of *LegoFuse* allow for fine-tuning all model parameters, including the ones of the initial modality-specific encoders.

## 4 EXPERIMENTAL SETUP

**Datasets.** We evaluate MM-Lego (*LegoMerge* and *LegoFuse*) and its components (*LegoBlock*) on seven multimodal medical datasets from three separate studies: The Cancer Genome Atlas (TCGA) (Institute, 2006), Medical Information Mart for Intensive Care (MIMIC) (Johnson et al., 2016) and the International Skin Imaging Collaboration (ISIC)) (Collaboration, 2020). The TCGA study includes data from four cancer subtypes (BLCA, BRCA, KIRP, UCEC) across four different modalities: digitalised pathology (whole slide image data), gene expression (continuous variables, tabular data), copy number variations (categorical variables, tabular data), and mutations (binary variables, tabular data). The MIMIC study includes intensive care data captured in two distinct modalities of continuous and time-series data. The ISIC study comprises image and time-series data pertaining to clinical data from melanoma patients. The seven tasks shown in our results correspond to survival analysis tasks on four TCGA sites (BLCA, BRCA, KIRP, UCEC), classification tasks on two variants of MIMIC (disease classification (ICD9) and patient in-hospital mortality (MORT)), and predicting melanoma for the ISIC patients. Full details on the datasets and their pre-processing steps can be found in Appendix C, and details on task losses and evaluation metrics are in Appendix D.

**Baselines.** For all experiments, we compare *LegoMerge* and *LegoFuse* to several uni- and multimodal baselines to evaluate their performance. For all tabular modalities, we use a self-normalising network (Klambauer et al., 2017) due to its performance and regularisation mechanisms suitable for high-dimensional tabular data. For the image and time series modalities, we use an attention-based Multiple Instance Learning (AMIL) (Shao et al., 2021). Across all modalities, we benchmark two related iterative-learning architectures: MultiModN (Swamy et al., 2023) and Perceiver (Jaegle et al., 2022), which generally show strong performance across a wide range of unimodal tasks. In terms of specific multimodal baselines, we use two late fusion combinations of SNN+AMIL, namely concatenation of their final latent representation and bi-linear fusion (Li et al., 2022). For the Perceiver, we use the same multimodal setup as suggested in the original paper, that is, concatenation of modalities before passing them into the model. We use two additional domain-specific multimodal baselines: the Hybrid Early-Fusion Attention Learning Network (HEALNet) (Hemker et al., 2024), which is using an end-to-end trained iterative cross-attention architecture, and the Multimodal Co-Attention Transformer (MCAT) (Chen et al., 2021), which is using the tabular (1D) modality as context for the imaging (2D) modality.

**Implementation Details.** For each experiment and dataset, we perform a 5-fold repeated random sub-sampling with a 70-15-15 train-test-validation split. We re-ran all of the baseline models in this paper using their open-source code to ensure that no performance differences are caused by different task setups, losses, or training splits. We ran a brief Bayesian Hyperparameter search (Biases, 2024) for key parameters of each model (learning rate, decay, schedulers, dropout, layer dimensions). The experiments were run on a single Nvidia A100 80GB GPU on a Ubuntu 22.04 virtual machine. The

code implementation for MM-Lego is available at `https://github.com/konst-int-i/mm-lego`.

## 5 RESULTS

Table 1: Mean and standard deviation of unimodal and multimodal task performance, showing the concordance Index (survival analysis tasks) and AUC (classification tasks) on 5 random sub-sampling folds with the **best** and **second-best** models highlighted. *LegoMerge* achieves competitive performance with multimodal baselines across all tasks without end-to-end training or any fine-tuning. *LegoFuse* achieves top-2 performance for all datasets and has the highest performance amongst all benchmarked models in five out of seven datasets. CC and BL denote the monolithic fusion operators $\psi(\cdot)$ of concatenation and bilinear fusion respectively. Modalities: img=image; ts=time series; mut=mutations (binary); cnv=copy number variations (categorical); rna=gene expressions (continuous); tab=other tabular

| | BLCA | BRCA | KIRP | UCEC | ICD9 | MORT | ISIC |
|---|---|---|---|---|---|---|---|
| *Samples* | n=436 | N=1021 | n=284 | n=538 | n=32616 | n=32616 | n=2875 |
| Modalities | img, mut, cnv, rna | img, mut, cnv, rna | img, mut, cnv, rna | img, mut, cnv, rna | tab, ts | tab, ts | tab, img |
| *Metric* | c-Index | c-Index | c-Index | c-Index | AUC | Macro AUC | AUC |
| **Unimodal (Tabular)** | | | | | | | |
| SNN | $0.689_{\pm0.012}$ | $0.544_{\pm0.020}$ | $0.798_{\pm0.035}$ | $0.589_{\pm0.057}$ | $0.731_{\pm0.023}$ | $0.634_{\pm0.020}$ | $0.507_{\pm0.005}$ |
| MultiModN | $0.500_{\pm0.000}$ | $0.500_{\pm0.000}$ | $0.525_{\pm0.140}$ | $0.500_{\pm0.000}$ | $0.500_{\pm0.000}$ | $0.500_{\pm0.000}$ | $0.500_{\pm0.000}$ |
| Perceiver | $0.686_{\pm0.009}$ | $0.557_{\pm0.016}$ | $0.836_{\pm0.053}$ | $0.615_{\pm0.035}$ | $0.629_{\pm0.023}$ | $0.658_{\pm0.000}$ | $0.840_{\pm0.084}$ |
| *LegoBlock* | $0.681_{\pm0.015}$ | $0.591_{\pm0.021}$ | $0.840_{\pm0.135}$ | $0.615_{\pm0.031}$ | $0.645_{\pm0.017}$ | $0.619_{\pm0.028}$ | $0.668_{\pm0.141}$ |
| **Unimodal (Image/Time-series)** | | | | | | | |
| ABMIL | $0.591_{\pm0.057}$ | $0.610_{\pm0.093}$ | $0.741_{\pm0.080}$ | $0.558_{\pm0.040}$ | $0.614_{\pm0.025}$ | $0.691_{\pm0.014}$ | $0.500_{\pm0.000}$ |
| MultiModN | $0.520_{\pm0.022}$ | $0.527_{\pm0.150}$ | $0.570_{\pm0.156}$ | $0.564_{\pm0.097}$ | $0.500_{\pm0.000}$ | $0.544_{\pm0.033}$ | $0.500_{\pm0.000}$ |
| Perceiver | $0.532_{\pm0.027}$ | $0.604_{\pm0.064}$ | $0.716_{\pm0.063}$ | $0.534_{\pm0.106}$ | $0.700_{\pm0.013}$ | $0.715_{\pm0.016}$ | $0.719_{\pm0.050}$ |
| *LegoBlock* | $0.568_{\pm0.029}$ | $0.533_{\pm0.000}$ | $0.630_{\pm0.182}$ | $0.565_{\pm0.069}$ | $0.643_{\pm0.013}$ | $0.711_{\pm0.008}$ | $0.706_{\pm0.147}$ |
| **Multimodal** | | | | | | | |
| *LegoMerge* | $0.701_{\pm0.021}$ | $0.601_{\pm0.025}$ | $0.825_{\pm0.114}$ | $0.625_{\pm0.080}$ | $0.684_{\pm0.015}$ | $0.751_{\pm0.027}$ | $0.721_{\pm0.143}$ |
| *Uplift (Merge vs. best Block)* | 2.9% | 1.7% | -1.8% | 1.6% | 5.7% | 5.3% | 2.1% |
| SNN + ABMIL (CC, Late) | $0.561_{\pm0.000}$ | $0.541_{\pm0.104}$ | $0.841_{\pm0.128}$ | $0.601_{\pm0.018}$ | $0.628_{\pm0.020}$ | $0.617_{\pm0.015}$ | $0.661_{\pm0.196}$ |
| SNN + ABMIL (BL, Late) | $0.622_{\pm0.054}$ | $0.557_{\pm0.089}$ | $0.811_{\pm0.108}$ | $0.666_{\pm0.031}$ | $0.500_{\pm0.000}$ | $0.500_{\pm0.001}$ | $0.501_{\pm0.002}$ |
| Perceiver (CC, Early) | $0.547_{\pm0.060}$ | $0.561_{\pm0.105}$ | $0.692_{\pm0.000}$ | $0.548_{\pm0.000}$ | $0.733_{\pm0.028}$ | $0.723_{\pm0.015}$ | $0.721_{\pm0.198}$ |
| MultiModN (Inter.) | $0.524_{\pm0.018}$ | $0.500_{\pm0.000}$ | $0.602_{\pm0.076}$ | $0.512_{\pm0.008}$ | $0.500_{\pm0.000}$ | $0.500_{\pm0.000}$ | $0.500_{\pm0.000}$ |
| MCAT (Inter.) | $0.702_{\pm0.032}$ | $0.564_{\pm0.000}$ | $0.823_{\pm0.076}$ | $0.633_{\pm0.068}$ | $0.500_{\pm0.000}$ | $0.500_{\pm0.000}$ | $0.627_{\pm0.059}$ |
| HEALNet (Inter.) | $0.714_{\pm0.025}$ | $0.618_{\pm0.063}$ | $0.842_{\pm0.063}$ | $0.594_{\pm0.023}$ | $0.767_{\pm0.022}$ | $0.748_{\pm0.009}$ | $0.639_{\pm0.09}$ |
| *LegoFuse, w/ 2 epochs* | $0.734_{\pm0.032}$ | $0.626_{\pm0.046}$ | $0.863_{\pm0.112}$ | $0.634_{\pm0.010}$ | $0.771_{\pm0.020}$ | $0.759_{\pm0.041}$ | $0.701_{\pm0.023}$ |

The results across the three prediction tasks (survival analysis, multi-class, and binary classification) are summarised in Table 1, showing the mean and standard deviation of the task-relevant performance metric across the 5 random sub-sampling folds. We compare our baselines to: 1) *LegoMerge*, which is taking two *LegoBlocks*, trained unimodally, and merges them without fine-tuning, and 2) *LegoFuse*,

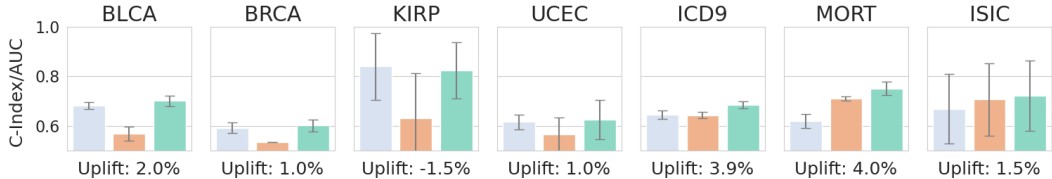

Figure 3: Mean task performance (concordance Index/AUC) of **LegoBlock (Tabular)**, **LegoBlock (Image/Time Series)** and **LegoMerge**, showing the increase in task performance by applying a multimodal model merge *without any fine-tuning*. Our proposed multimodal model merge shows a positive performance improvement on 6 out of 7 datasets.

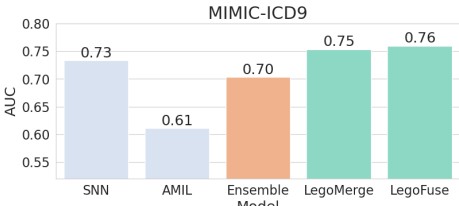 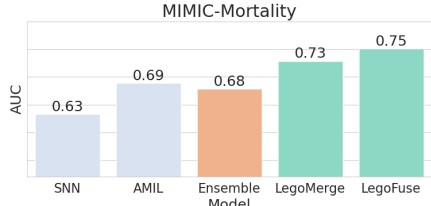

Figure 4: AUC performance on the MIMIC dataset when merging existing encoders (SNN for tabular, AMIL for Time Series) using **LegoMerge** and **LegoFuse**. Our multimodal model merge shows much better performance than using an **ensemble**, exhibiting the performance gains, at no additional costs, through the merge even prior to fine-tuning in **LegoFuse**.

which is taking the same blocks and fine-tuning them for two epochs. Note that, we did not see a significant performance difference between the "stack" and "weave" variants of *LegoFuse*, therefore the reported results in Table 1 correspond to *stacked* blocks. Across all datasets, *LegoFuse* is within the top 2 performers of all uni- and multimodal datasets with only two epochs of fine-tuning.

Both *LegoFuse* and *LegoMerge* generalise well across domains (pathology, clinical care, dermatology), which some unimodal and multimodal baselines struggle with and heavily overfit (AUC of 0.5). Despite never seeing a single multimodal training step, *LegoMerge* achieves strong results, closely matching the performance of the best baselines in six out of seven datasets. We note that the performance stability across folds is in line with or better than the multimodal baselines for all datasets except TCGA-KIRP, where we have significantly larger standard deviations. We believe that this instability is caused by the relatively low sample size of this dataset, and posit that this could be alleviated by pre-training *LegoBlocks* on other TCGA datasets before fine-tuning them on KIRP. The SIIM-ISIC dataset exhibits a case of multimodal collapse (also referred to as dominance), where all multimodal models struggle to effectively take advantage of both modalities.

We assess the gains afforded by *LegoMerge* in Figure 3, where we use *LegoMerge* to merge *LegoBlock* for tabular data with either *LegoBlock* for imaging or *LegoBlock* for times series data (depending on the dataset). We compare the performance of the unimodal *LegoBlocks* with the resulting multimodal *LegoMerge* and find a performance uplift gained from the merge. Namely, the harmonic mean (Equation 4) of each block's latent states coupled with spherical linear interpolation of the weights and biases in the task heads leads to a better performance than for either unimodal block in 6 out of 7 datasets. Again, the outlier to this trend is the dataset with the lowest sample size (KIRP) where *LegoMerge* generally struggles for stability. As outlined in Figure 1, the blocks can either be trained as models from scratch or can be used as a wrapper for a unimodal encoder. This is demonstrated in Figure 4, which shows the performance of the SNN and AMIL, compared to three multimodal combinations of the two: 1) a naive ensemble that is taking the average logits of each encoder, 2) *LegoMerge* applied after wrapping each encoder in *LegoBlock*, and 3) *LegoFuse* with limited fine-tuning. For both MIMIC tasks, we can see that *LegoMerge* beats the ensemble by 7.1% and 7.3% on the MIMIC disease classification (ICD9) and mortality prediction (MORT) respectively. Moreover, *LegoFuse* improves the performance even more over all other models by a further 1-3%.

## 6 DISCUSSION

In this work, we introduce two novel approaches for both multimodal model merging (*LegoMerge*) and fusion (*LegoFuse*) to address some common limitations of existing methods for multimodal modelling. We outline five desirable criteria of multimodal models for biomedical data in Tabel 2 to highlight scenarios in which we believe our introduced methods to be highly beneficial.

**Performance without end-to-end training.** With the increasing volume, complexity and diversity of collected biomedical data, (re)training multimodal models from scratch becomes more expensive, unsustainable, and even infeasible in the long run. Similar to trends seen in large language models, providing access to fine-tuning and merging frameworks can help to make established state-of-the-art unimodal models more accessible to tailor for highly specific applications, as it is commonly a requirement in medicine. The results in Table 1 show that we can achieve state-of-the-art performance

by combining existing, pre-trained unimodal models and fine-tuning them accordingly: *LegoFuse* outperforms end-to-end trained baselines with as little as 2 epochs of fine-tuning. We anticipate that *LegoMerge*, where no multimodal supervised data is required, and *LegoFuse*, where only a limited number of supervised paired samples are required, can aid in fields where paired observations are scarce. These scenarios are frequent and very realistic, from studies on rare diseases to data gathered from clinics that do not have access to sophisticated data acquisition technologies. In such cases, training large encoders wrapped in *LegoBlocks* on similar domains, and then fine-tuning them on a small amount of supervised paired samples may present a feasible path for domain adaptation for multimodal models. A further advantage of our modular approach is that additional data (from either the same or novel modalities) can be added at different points in time. For example, if a large bimodal model was already trained and a new modality becomes available later on, MM-Lego can be readily applied to extend such model without training a new model, saving time, costs and energy.

**Missing modalities and modality imbalance.** The results in Section 5 support our hypothesis that we can build performant multimodal models without having perfectly paired training data. This is evident by the highly competitive performance of *LegoMerge* in comparison with existing multimodal baselines (Table 1). Paired data requirements can be problematic when modalities are missing for some of the samples, leading to a data-performance trade-off: one can either attempt to use all the available data and impute the missing values, which introduces noise, or can take the intersection of samples with available data modalities, which may dramatically reduce the sample size available for training.

MM-Lego overcomes this data-performance trade-off as it can be effectively trained on non-overlapping training sets. This is further supported by our experiments (see Appendix G), where we found that the performance of MM-Lego remains stable even when trained with completely non-overlapping (symmetrically different) sets of samples. In contrast, such a training scenario is infeasible for any other current end-to-end model setup. Moreover, MM-Lego can effectively handle *missing modalities*. During training, value imputation is not an issue for MM-Lego, since we can train *LegoBlocks* independently, and subsequently combine them into a performant model, as shown in Figure 4. During inference, we can easily skip missing modalities, which is another benefit of making each block compatible with an iterative attention architecture (Equation 1). That is, while a monolithic fusion operator $\psi(\cdot)$ requires the entire set of unimodal encodings $\mathcal{H}$ to be present, MM-Lego's architecture can just query the elements in $\mathcal{B}$ for the available modalities. Finally, MM-Lego's modular design allows for handling high modality imbalance and one-to-many cardinalities ($|X^{(A)}| \neq |X^{(A)}|$). An example of such a case would be a dataset where $A$ are the histopathology slides (one per patient) and $B$ is the associated single-cell data (many per patient).

**Architecture agnostic.** A key limitation of existing model merging literature in multi-task learning is the assumption that the majority of the network topology between tasks is equivalent. While this is a feasible assumption for merging in multi-task learning, the data heterogeneity limits its application in multimodal settings. Therefore, the design of *LegoBlock* is sufficiently permissive to use any unimodal encoder as part of this framework, while complying with the necessary architectural assumptions required for model merging. Our results in Table 1 and Figure 4 support this by showing that any unimodal encoder (SNNs and AMIL in this example) can be wrapped in a *LegoBlock* without any practical loss in performance, whilst making them capable for further merging and/or fusion.

Table 2: Comparison of desirable requirements of multimodal systems for clinical practice. *LegoMerge* and *LegoFuse* present two alternatives to existing, end-to-end trained, multimodal fusion approaches (early, intermediate, and late fusion) and multi-task merge methods applied to multimodal settings. ✓: meets requirement, (✓): some approaches meet requirement, ✗: fails requirement.

| *Criteria/Method* | Fusion | | | Multi-task Merge | **LegoMerge** | **LegoFuse** |
|---|---|---|---|---|---|---|
| | Early | Intermediate | Late | | | |
| Learns cross-modal interactions | ✗ | ✓ | (✓) | ✗ | ✗ | ✓ |
| Architecture agnostic | ✓ | (✓) | ✓ | ✗ | ✓ | ✓ |
| Handles strong modality imbalance | ✗ | (✓) | ✗ | ✓ | ✓ | ✓ |
| Add modalities without re-training | ✗ | ✗ | ✗ | ✗ | ✓ | (✓) |

To the best of our knowledge, MM-Lego is the first general-purpose model merging framework for multimodal data outside of the vision & language domain.

**Low computational requirements**. Scaling to more than two modalities is increasingly important for multimodal models as more modalities are captured at scale: this was one of the main motivations for the modular design of MM-Lego. Many intermediate fusion approaches (Chen et al., 2021; Xu & Chen, 2023; Sundar et al., 2024; Zhang et al., 2024) are natively designed for two modalities, and centred around a cross-attention layer between the modalities with a time complexity of $\mathcal{O}(N^2)$. Scaling to more than two modalities requires calculating the modality-guided cross-attention for all unique pairwise combinations $\binom{M}{2} = \frac{M(M-1)}{2}$, which is $\mathcal{O}(M^2)$, leading to a total upper bound time complexity of $\mathcal{O}(M^2N^2)$. MM-Lego improves on this wrt. both $M$ and $N$. The time complexity of *LegoBlock* is bound by the cross-attention unit, which reduces the quadratic time complexity by using the latent $L \in R^{c \times d}$ as the query to $\mathcal{O}(dN)$ for latent dimensions $d$. The sequential design in Equation 1 ensures linear scaling wrt. the number of modalities, leading to a final time complexity of $\mathcal{O}(dMN)$. Beyond time complexity, another key benefit of MM-Lego is the number of training steps required at the given complexity. For the end-to-end baselines, we typically observe loss convergence around $\sim 10$ training epochs, while *LegoMerge* and *LegoFuse*, by design, require zero or as little as two training epochs, respectively. This results in a very low total train time, as shown in Table 3.

Table 3: Train time per epochs and total wall time for training on the TCGA-UCEC dataset on a single A100 80GB GPU. For encoders wrapped with a *LegoBlock* during training, *LegoMerge* achieves competitive performance despite requiring no additional training.

| Model | Time/epoch (s) | Train Time (s) |
|---|---|---|
| SNN+ABMIL (BL) | 5.7 | 102.6 |
| SNN+ABMIL(CC) | 4.7 | 84.6 |
| Perceiver | 9.6 | 172.8 |
| MultiModN | 3.2 | 57.6 |
| MCAT | 4.8 | 86.4 |
| HEALNet | 10.1 | 181.8 |
| **LegoMerge** | **0** | **0** |
| **LegoFuse** | **8.1** | **16.2** |

**Limitations.** We believe that our MM-Lego approach would benefit from further research of parameter-efficient ways to design the fusion layer within each *LegoBlock* (Equation 3), that is, efficiently encoding $h^{(A)}$ into the updated latent representation $(L_{t+1}^{\mathcal{F}})^r$. Whilst this is suitable in both our solutions and has proven to be effective in other work (Jaegle et al., 2022; Carreira et al., 2022; Hemker et al., 2024), finding a more parameter-efficient solution is desirable. Note that, employing the LegoBlock adapters requires minimal fitting. This can be achieved by either training it together with the modality-specific encoder or retrospectively, when applied to a pre-trained encoder (e.g., a large foundation model). In the latter case, this still requires a small amount of training (supervised) samples for LegoMerge to be effective. As such, MM-Lego would benefit from further research on self-supervised fitting of the adapters. Finally, while in this paper we limit our focus to multimodal problems from biomedical domains, MM-Lego is designed to be general-purpose and applicable to any multimodal tasks. Therefore, we leave it to future work to further demonstrate these properties for more (and other) data modalities and domains, including vision & text tasks.

## 7 CONCLUSION

We present MM-Lego, a general-purpose and modular learning framework to build performant multimodal models with minimal fine-tuning. To achieve this, we introduce three novelties. First, we introduce a wrapper for unimodal encoders (*LegoBlock*) that a) enforces shape consistency between modality representations, and b) harmonises the latent representations in the frequency-domain to enable model merging with little signal interference. Second, we introduce the first multimodal model merge framework (*LegoMerge*) that goes beyond vision & language modalities and outperforms many unimodal baselines without seeing a single multimodal supervised training sample. Third, we show that these building blocks can be combined to construct a fusion model that achieves state-of-the-art performance with only a few epochs of fine-tuning (*LegoFuse*).

ACKNOWLEDGMENTS

The authors would like to thank Philip Schouten for his insightful clinical feedback. KH acknowledges support from the Gates Cambridge Trust via the Gates Cambridge Scholarship. NS and MJ acknowledge the support of the U.S. Army Medical Research and Development Command of the Department of Defense; through the FY22 Breast Cancer Research Program of the Congressionally Directed Medical Research Programs, Clinical Research Extension Award GRANT13769713. Opinions, interpretations, conclusions, and recommendations are those of the authors and are not necessarily endorsed by the Department of Defense.

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

APPENDIX

## A  NOTATION

**Objects.**

- $X^{(A)}$: matrix corresponding to modality A
- $\mathbf{x}^{(A)}$: a vector in $X^{(A)}$ (e.g., a sample of modality A)
- $\mathbf{X}^{(A)}_{i,j,k}$: elements of matrix $X^{(A)}$ at row $i$, column $j$, channel $k$, assuming $X^{(A)} \in \mathbb{R}^{I \times J \times K}$ where $1 \leq i \leq I, 1 \leq j \leq J, 1 \leq k \leq K$
- $\mathbf{X}^{(\mathcal{M})} = \bigcup_{m \in \mathcal{M}} X^{(m)}$: multimodal dataset
- $\mathbf{y} \in \mathcal{Y} = \bigcup_{t \in \mathcal{T}} \mathbf{y}^{(t)}$: set of task labels for all available tasks $\mathcal{T}$
- $\mathbf{y}^{(T_1)}$: task labels for task $T_1$

**Sets.**

- $\mathcal{M}$: set of modalities
- $\mathcal{T}$: set of tasks
- $\mathcal{Y}$: set of task-specific heads
- $\mathcal{G} = \{g_m : m \to \mathbf{h}^{(m)} \mid m \in \mathcal{M}\}$: set of modality-specific encoders
- $\mathcal{H}_{\mathbf{y}} = \{g_m(m, \mathbf{y}) \mid m \in \mathcal{M}\}$: set of task- and modality-specific embeddings
- $\mathcal{B} = \{\psi_m : (g_m(X^{(m)}), L_s^{(m)}) \to L_{s+1}^{(m)} \mid s \in S, m \in \mathcal{M}\}$: set of *LegoBlocks*

**Functions and Operators.**

- $g_m(\cdot)$: modality-specific encoder
- $\psi(\cdot)$: fusion operator (monolithic)
- $\psi_m(\cdot)$: modality-specific latent update
- $\mathcal{F}$: Fourier transform
- $\mathcal{F}^{-1}$: Inverse Fourier transform

## B  PROPERTIES OF FREQUENCY-DOMAIN REPRESENTATIONS

In this section, we further elaborate on the desirable properties of frequency-domain representations that motivate *MM-Lego* design.

- **Signal preserving**: The general motivation for signal-preserving aggregation for multimodal machine learning has been extensively studied in the context of fusion methods. For example, bilinear pooling of two latent vectors $h_1 \in \mathbb{R}^n$ , $h_2 \in \mathbb{R}^m$ and resulting tensor $h_{12} \in \mathbb{R}^d$ ($y = h_1^T A h_2 + b$) with a learnable parameter $A \in \mathbb{R}^{n \times m \times d}$ follows the same motivation. Several related works in different domains and modalities [1, 2, 3] emphasise the importance of fusion or aggregation methods that avoid signal interference, where meaningful patterns or complementary information from individual modalities may be attenuated, distorted, or lost due to naive aggregation approaches such as averaging or summation. With the same motivation in mind, we take advantage of the Fourier transform's orthogonal properties, as after the transform each frequency component (represented by sine and cosine waves) is orthogonal to the others, such that $\int_{-\infty}^{\infty} sin(\omega_1 h_1) cos(\omega_2 h_1) dh = 0$ for angular frequencies $\omega_1 \neq \omega_2$ [4]. This means that the contribution of each frequency is independent of others and there is no overlap between them. It follows that if we take the harmonic mean of two fourier-transformed latents $\mathcal{F}_1(\omega)$ and $\mathcal{F}_2(\omega)$ as $H(w) = \frac{2\mathcal{F}_1(\omega)\mathcal{F}_2(\omega)}{\mathcal{F}_1(\omega)+\mathcal{F}_2(\omega)}$, the harmonic mean aggregates each $\omega$ in a localised manner. This means that each frequency component in $\mathcal{F}_1(\omega)$ interacts only with the corresponding frequency component in $\mathcal{F}_2(\omega)$, i.e., without interference from other frequencies.

- **Distance preserving**: Parseval's Theorem shows that the Fourier transform is a unitary operator, meaning that the sum or integral of the square of a function is equal to the sum or square of its transform [4,5]. As such, the distances between two signals are the same between the transformed and untransformed representations. - Concretely, the theorem states that the energy of a signal is preserved in both the time domain and frequency domain, where its energy is measured as the integral of the function.
  - Formally $\int_{-\infty}^{\infty} |f(h)|^2 dh = \int_{-\infty}^{\infty} |\mathcal{F}(f)(\omega)|^2 d\omega$ for latent signal $f(h)$ and its fourier transform $\mathcal{F}$.
  - The Euclidean distance between two signals $f_1(h)$ and $f_2(h)$ in the spatial domain is: $||f_1 - f_2|| = \sqrt{\int_{-\infty}^{\infty} |f_1(h) - f_2(h)|^2 dh}$
  - The Euclidean distance between the Fourier transforms of two signals is $||\mathcal{F}(f_1) - \mathcal{F}(f_2)|| = \sqrt{\int_{-\infty}^{\infty} |\mathcal{F}(f_1)(\omega) - \mathcal{F}(f_2)(\omega)|^2 d\omega}$
  - From Parseval's Theorem, it follows that $||f_1 - f_2|| = ||\mathcal{F}(f_1) - \mathcal{F}(f_2)||$.
  - This distance-preserving capability is beneficial in designing loss functions in a multimodal setting.

- **Invertible**: The Fourier transform is not an idempotent function but periodic with a period of 4, i.e., $\mathcal{F}^4(f) = f$. This would prevent the iterative architecture outlined in Section 3 from working since a repeat application of the transform would lead to different representations. Meanwhile, the Fourier Inversion Theorem [6] shows that we can invert the frequency-domain representation to its original function without the 4x repeat application, making it suitable for the chosen iterative architecture.

- **Efficient**: The Fast Fourier Transform (FFT) has a time complexity of $\mathcal{O}(nlog(n))$ making it scalable to very large datasets.

## C   DATASETS

We evaluate MM-Lego (*LegoMerge* and *LegoFuse*) and its components (*LegoBlock*) on seven multi-modal medical datasets covering three separate modalities (images, tabular, time series) from three separate sources: histopathology (The Cancer Genome Atlas (TCGA)) Institute (2006), intensive care data (Medical Information Mart for Intensive Care (MIMIC)) Johnson et al. (2016), and skin imaging (Society for Imaging Informatics in Medicine & International Skin Imaging Collaboration (SIIM-ISIC)) Collaboration (2020).

**TCGA**: Some of the results shown in this paper here are based upon data generated by the TCGA Research Network: `https://www.cancer.gov/tcga`. The Cancer Genome Atlas (TCGA) is an open-source genomics program run by the United State National Cancer Institute (NCI) and National Human Genome Research Institute, containing a total of 2.5 petabyts of genomic, epigenomic, transcriptomic, and proteomic data. We predict survival of right-censored patients based on the high-resolution histopathology slides ($\sim 80,000 \times 80,000$ pixels) and multi-omic data (gene expressions, copy number variations and gene mutations) captured from bulk sequencing in a tabular format. We train on four separate cancer cohorts with multimodal data available: Urorethelial Bladder Carcinoma (BLCA, $n = 436$), Breast Invasive Carcinoma (BRCA, $n = 1021$), Kidney Renal Papillary Cell Carcinoma (KIRP, $n = 284$), and Uterine Corpus Endometrical Carcinoma (UCEC, $n = 538$).

We use the following encoders for each modality in the TCGA datasets:

- WSI Encoder
    - Sample Input: Raw image converted to a bag of patches, $d_{wsi} = n_{patches} \times 256 \times 256 \times 3$
    - Encoded input using ResNet50, pre-trained on Kather100k : $h_{wsi} = n_{patches} \times 2048$
- Mutation Encoder
    - Non-variable mutation genes were filtered (i.e., every sample contains mutation or none contains mutation)
    - Input: Raw input mutation data vector for cross-attention (Figure 2)
    - Encoder: none as the vector is already relatively small after ETL: $d_{mut} = h_{mut} = 21$
- Copy Number Variation Encoder
    - Non-variable copy number genes were filtered
    - Input: Copy number variations for each gene (categorical variable ranging -2 (deep deletion) to +2 (strong oncogenic amplification): $d_{cnv} = 1333$
    - Encoder: SNN ($1333 \to 512$): $h_{cnv} = 512$
- Gene Expression Encoder
    - Low variable genes were filtered
    - Input: log1p transformed bulk RNAseq gene expression: $d_{rna} = 1558$
    - Encoder: SNN($1558 \to 512$): $h_{rna} = 512$

**MIMIC-III**: We train models on two separate tasks: patient mortality (multi-class classification) and disease classification (ICD-9 codes), which we formulate as a binary classification task. We use both clinical variables and small time series data on various vital signs measured at 24 time steps. Both tasks have $n = 32616$ and the same feature set for different task labels.

**SIIM-ISIC**: Stems from the Society for Imaging Informatics in Medicine & International Skin Imaging Collaboration (SIIM-ISIC) melanoma classification Kaggle challenge Collaboration (2020), which contains both tabular data and images of skin leisures to be classified for melanoma patients. To account for class imbalance, we randomly downsampled the majority class to a 5:1 ratio for the class of interest (melanoma) to a sample size of $n = 2875$. All images were patched and encoded using the resnet50-kather100k for TCGA (ResNet pre-trained on a large histopathology patch collection) Pocock et al. (2022) and a regular ImageNet v2 pre-trained ResNet for the pictures of skin leisures. Both images (patch encodings) and times series were represented as 2D tensors, and the tabular clinical and multi-omic data as 1D tensors to pass into the modality-specific encoders $g(\cdot)$.

# D    LOSSES AND METRICS

The results report the (unseen) test set performance, by evaluating the concordance Index (c-Index) in the case of TCGA, AUC in the case of MIMIC-III-ICD9 and ISIC, and Macro-AUC ("one-vs-rest") for MIMIC-III-ICD9. As indicated in Figure 1 the output of each task head in $\mathcal{Y}$ are the logits with predictions for each class given the final Fourier-transformed latent state $y_l = f(L_T^{\mathcal{F}})$. Since TCGA is a survival prediction task with right-censored data, we have divided the survival period into four non-overlapping bins and use the logits of these bins to calculate the hazard ($y_h = \frac{1}{1e^{-y_l}}$) and survival ($y_s = \prod_1^k 1 - y_h$) respectively for $k$ bins. Given the hazards, censorship, and ground truth bins, we can calculate the negative log-likelihood loss from a proportional hazards model Zadeh & Schmid (2021) which is used as the survival loss. We evaluate the performance using the Concordance Index (c-Index), for which we determine the fraction of paired samples in which the prediction outcomes are concordant with the ground truth. As MIMIC and ISIC relate to classification tasks, we employ categorical cross-entropy loss for training. Note that both AUC and the c-Index have similar interpretations, therefore the values range between $[0.5 - 1]$.

# E  HYPERPARAMETERS

| Scope | Parameter | Value |
|---|---|---|
| Shared | Learning Rate | 0.003 |
| | Epochs | 40 |
| | Early Stopping Patience | 7 |
| | L1 Regularization | 0.0002 |
| | Batch size | 512 |
| | Optimizer | Adam |
| | LR Scheduler | ReduceLROnPlateau |
| MM-Lego | Tune Epochs | 2 |
| | Fuse Method | Stack |
| | Merge method | Harmonic |
| | Head method | SLERP |
| | Alpha | 0.5 |
| | Track imaginary | True |
| | Normalise | True |
| | Latent dims | 17 x 126 |
| | Depth | 4 |
| | Attention Dropout | 0.45 |
| | FCNN Dropout | 0.36 |
| MultiModN | Latent dims | 1000 |
| | Error penalty | 1 |
| | State change penalty | 0 |
| | Layer dims | 512, 256, 128, 64 |
| HEALNet | Depth | 2 |
| | Latent dims | 17x126 |
| | Attn Head dims | 64 |
| | Attention Dropout | 0.4 |
| | FCNN Dropout | 0.27 |
| AMIL | Layer sizes | 1024, 512, 256 |
| | Dropout | 0 |
| | Attention Dropout | 0.25 |
| SNN | Layer dims | 256 |
| | Depth | 4 |
| | Dropout | 0.25 |
| Perceiver | Depth | 2 |
| | Latent dims | 17x126 |
| | Attn Head dims | 64 |
| | Attention Dropout | 0.4 |
| | FCNN Dropout | 0.36 |
| MCAT | AMIL layers | 1024, 512, 256 |
| | AMIL dropout | 0 |
| | Attention Dropout | 0.25 |
| | SNN Layer dims | 256 |
| | SNN Depth | 4 |
| | SNN Dropout | 0.25 |

# F  SIGNAL INTERFERENCE ON LATENT VARIABLES

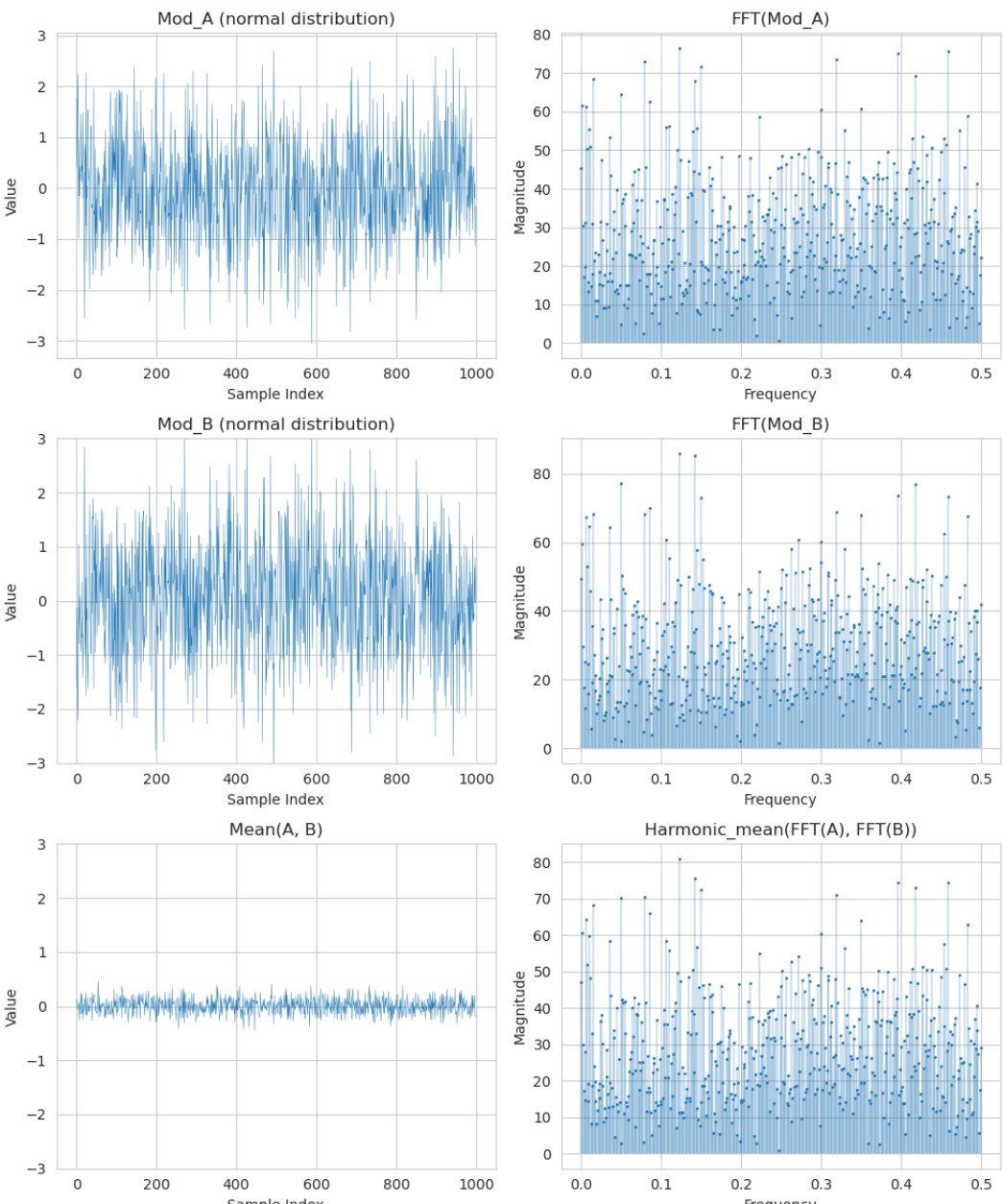

Figure 5: Example of signal interference on a random normal latent variable and its additive inverse variable with some added noise, showcasing a severe case of signal interference where nearly all signal cancels out. We can see that the fourier-transformed data does not suffer this problem when we apply the harmonic mean. This is a key reason for the choice of model merging architecture.

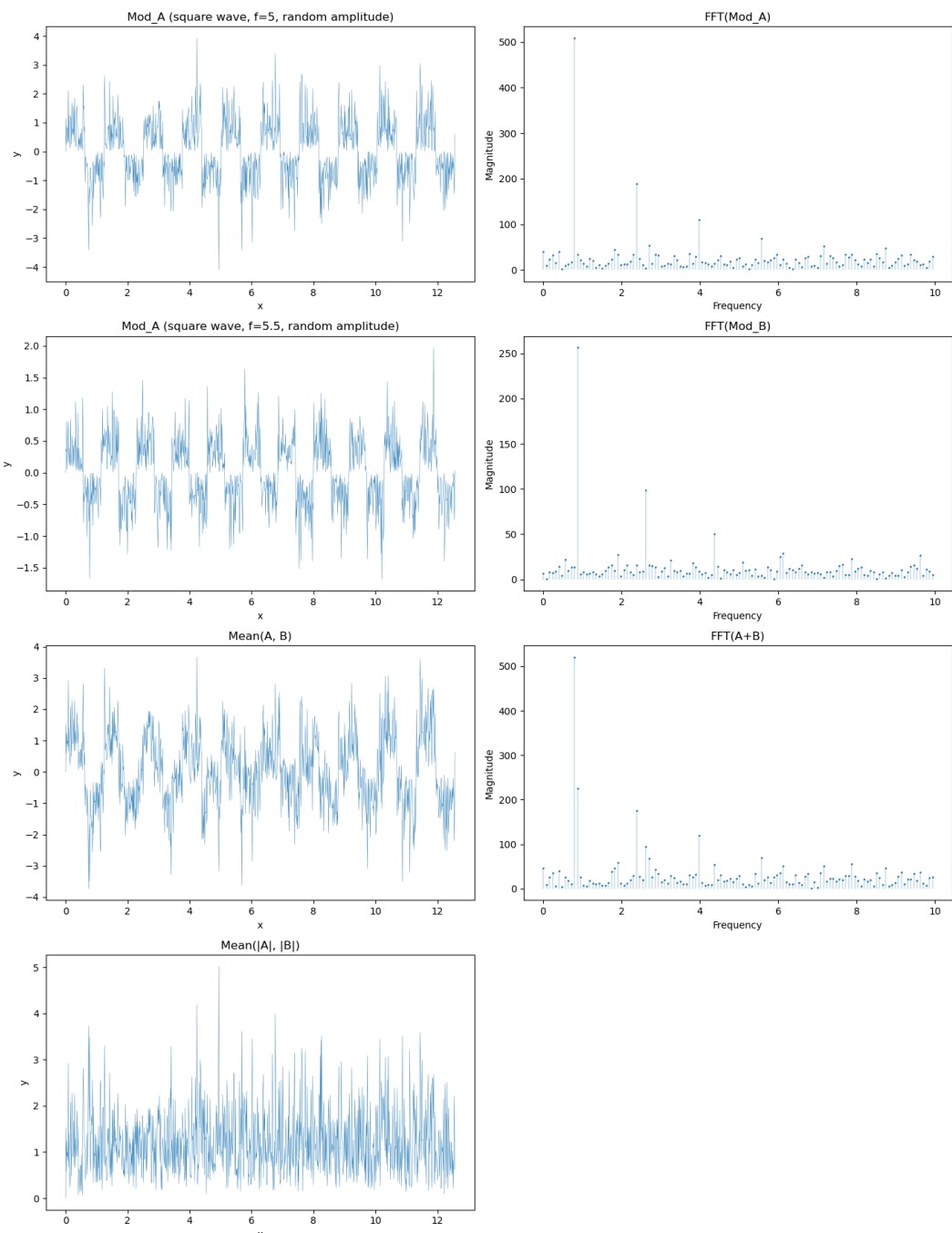

Figure 6: The argument against Fig. 5 would be to use absolute or only positive values. This example shows that this logic can also be flawed. We demonstrate this using a squarewave function with a frequency offset beteen $Mod_A$ and $Mod_B$ and a scaled amplitude by a normal distribution. We can see that the mean of the regular and the absolute values suffers some signal interference while the FFT aggregation does not.

# G TRAINING ON UNPAIRED DATA

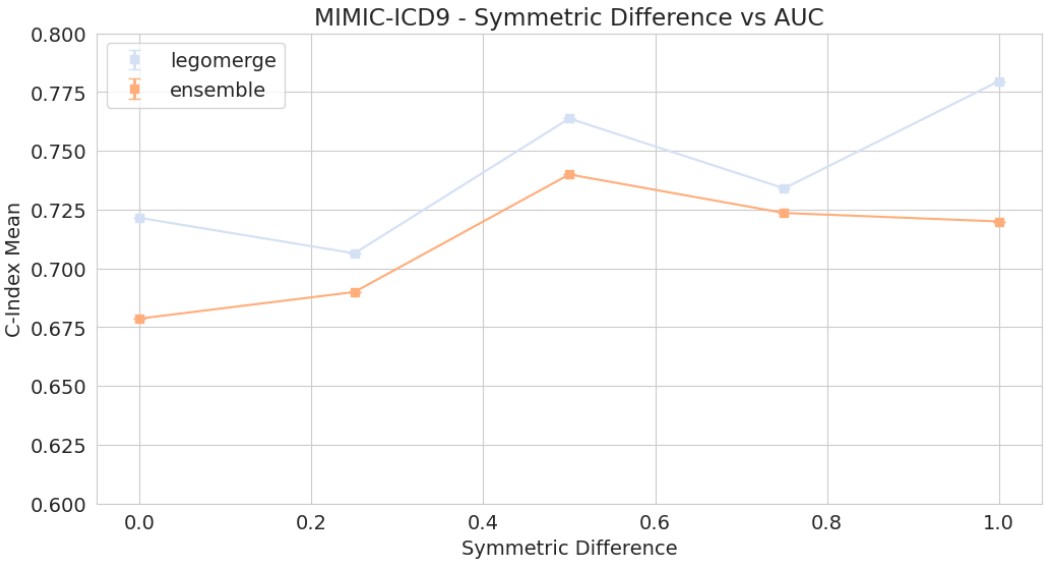

Figure 7: Test performance of *LegoMerge* (SNN+AMIL) compared to the SNN-AMIL ensemble when training on different levels of overlapping samples between the modalities. A symmetric difference of 1 means no overlap between the samples, 0 being perfect overlap. We selected N=10,000 MIMIC examples for this experiment.

