# OpenReview forum: "Multimodal Lego: Model Merging and Fine-Tuning Across Topologies and Modalities in Biomedicine"
_ICLR.cc/2025/Conference — ICLR 2025 Poster_

### Official Review · Reviewer_fqpF · 2024-11-01

**Soundness:** 3
**Presentation:** 2
**Contribution:** 2
**Rating:** 5
**Confidence:** 4

**Summary:**

This paper presents MM-Lego, a general-purpose and modular learning framework to build performant multimodal models with minimal fine-tuning.  They introduce a wrapper for unimodal encoders to enforce shape consistency between modality representations and also harmonize the latent representations in the frequency domain to enable model merging with little signal interference. This paper conduct experiments on biomedical datasets and shows good results.

**Strengths:**

1. The study problem is interesting and useful. How to fuse the feature/prediction from multimodal data better.
2. The proposed framework demonstrate good results on the datasets.

**Weaknesses:**

In general, the presentation of this paper needs improvement, and the experimental section should be enhanced.

1. The motivation for some key parts is not clearly articulated. For example, the authors mention "topological equivalence" and "topology agnostic." However, it is unclear why these properties are necessary for the problem being addressed.

2. The proposed framework falls under the category of "model merging." However, I am uncertain if this classification is accurate. Essentially, the authors aim to explore how to enhance the fusion of different modalities.

3. The key technical advancement appears to be the feature fusion conducted in the frequency domain. Again, the motivation for this approach is not well presented. Why is this method necessary?

4. The authors claim that the proposed framework is scalable, particularly regarding the number of modalities. However, in most of the experiments, they only utilize two modalities. This claim requires more experimental results to substantiate it.

5. It is unclear how the proposed framework addresses the issue of missing modality data, which is an important challenge in the field.

6. The authors should elaborate more on the experimental settings and the methods used for comparison. For instance, it would be helpful to know if they employed the same feature backbone across experiments.

**Questions:**

Please see the weakness.

---

> ### Author Response · Authors · 2024-11-21
>
> Thank you for your thoughtful comments and suggestions for improvement. In the following, we provide detailed answers to your comments with additional explanations of the raised issues. We also include these clarifications and additions in the manuscript.
>
> **[Q1] Clarifications on "topological equivalence" and "topology agnostic."**
>
> In this paper, we focus on multimodal learning from multiple *heterogeneous* modalities. Most current state-of-the-art fusion methods, that require end-to-end training, can be very computationally demanding and often do not scale beyond two modalities. Moreover, in order to train end-to-end fusion methods, we usually require a large amount of paired data for training (e.g., a vision & language model, where each image needs to be paired with a text string to learn a meaningful representation). In many multimodal learning problems for biology and medicine, which this paper focuses on, large amounts of paired data are relatively scarce. At the same time,  single modalities are often widely available (just not paired).
>
> Model merging, on the other hand, allows for combining well-performing unimodal models trained in isolation, eliminating the need for paired data. Most work in this field focuses on model merging in *multi-task* settings, i.e., where you have two identical model architectures (topologies) trained on separate tasks and combine them to obtain a single model instance that performs well on both tasks [1, 2]. For equivalent topologies, this process is relatively straightforward as you have a 1:1 mapping of layers and their corresponding weights between the two task-specific models. However, in multimodal settings, we often *cannot* use equivalent architectures e.g., with imaging and tabular data as their architectures need to handle different heuristics and tensor shapes. Therefore, in multi-modal learning, having a *topology agnostic* model merging approach w.r.t the modality-specific encoders is critical to enable model merging.
>
> *This is the main contribution of our work*: MM-Lego introduces modality-specific adapters that enable model merging even if the architecture is not equivalent — allowing users to combine any set of unimodal models (including current state-of-the-art foundation models) into a multimodal model with very minimal fine-tuning.
>
> **[Q2] Clarifications on the taxonomy of "model merging"**
>
> We propose two variants of MM-Lego: LegoFuse and LegoMerge. *LegoFuse* is indeed closer to a fusion method, where we aim at combining models before further fine-tuning them to learn cross-modal interactions on a limited amount of paired data. LegoMerge, on the other hand, is a model merging technique whose definition falls in line with other model merging methods, similar to that in multi-task literature — that is, the model weights are merged and inference is done without additional training. As mentioned in our response to Q1, what sets LegoMerge apart from other merging techniques is that it does not require equivalent architectures between the individual models being merged. As such, it can be readily applied in any scenario, be it merging multimodal from heterogeneous modalities (e.g., image + tabular) or merging multiple different models on a single dataset (e.g., combining multiple vision foundation models trained on separate distributions).

---

> ### Author Response · Authors · 2024-11-21
>
> **[Q3] On the motivation of using frequency-domain representations**
>
> We would like to further elaborate on the motivation to use frequency-domain representations which the paper touches upon in L203-221. We will add a more detailed discussion of the motivation and need for frequency-domain representations in the paper appendix.
>
> - Signal preserving: The general motivation for signal-preserving aggregation for multimodal machine learning has been extensively studied in the context of fusion methods. Several related works in different domains and modalities [3, 4, 5] emphasise the importance of fusion or aggregation methods that avoid signal interference, where meaningful patterns or complementary information from individual modalities may be attenuated, distorted, or lost due to naive aggregation approaches such as averaging or summation. For example, bilinear pooling [3] of two latent vectors $h_1 \in \mathbb{R}^n$ , $h_2 \in \mathbb{R}^m$ and resulting tensor $h_{12} \in \mathbb{R}^d$ ($y=h_1^T A h_2 + b$) with a learnable parameter $A \in \mathbb{R}^{n \times m \times d}$ follows the same motivation. With the same motivation in mind, we take advantage of the Fourier transform’s orthogonal properties, as after the transform each frequency component (represented by sine and cosine waves) is orthogonal to the others, such that $\int_{-\infty}^{\infty} sin(\omega_1h_1) cos(\omega_2h_1) dh = 0$ for angular frequencies $\omega_1 \neq \omega_2$ [4]. This means that the contribution of each frequency is independent of others and there is no overlap between them. It follows that if we take the harmonic mean of two fourier-transformed latents $\mathcal{F}_1(\omega)$ and $\mathcal{F}_2(\omega)$ as $H(w) = \frac{2\mathcal{F}_1(\omega)\mathcal{F}_2(\omega)}{\mathcal{F}_1(\omega)+\mathcal{F}_2(\omega)}$, the harmonic mean aggregates each $\omega$ in a localised manner. This means that each frequency component in $\mathcal{F}_1(\omega)$ interacts only with the corresponding frequency component in $\mathcal{F}_2(\omega)$, i.e., without interference from other frequencies.
> - Distance preserving:  Parseval’s Theorem shows that the Fourier transform is a unitary operator, meaning that the sum or integral of the square of a function is equal to the sum or square of its transform [4,5]. As such, the distances between two signals are the same between the transformed and untransformed representations.
>     - Concretely, the theorem states that the energy of a signal is preserved in both the time domain and frequency domain, where its energy is measured as the integral of the function. Formally $\int_{-\infty}^{\infty} |f(h)|^2dh = \int_{-\infty}^{\infty} |\mathcal{F}(f)(\omega)|^2 d \omega$ for latent signal $f(h)$ and its fourier transform $\mathcal{F}$.
>     - The Euclidean distance between two signals  $f_1(h)$ and $f_2(h)$ in the spatial domain is: $||f_1-f_2||  = \sqrt{\int_{-\infty}^{\infty}|f_1(h) - f_2(h)|^2 dh }$
>     - The Euclidean distance between the Fourier transforms of two signals is:
>     - $||\mathcal{F}(f_1) - \mathcal{F}(f_2)|| = \sqrt{\int_{-\infty}^{\infty} | \mathcal{F}(f_1)(\omega) - \mathcal{F}(f_2)(\omega)|^2 d\omega}$
>     - From Parseval’s Theorem, it follows that $||f_1 - f_2 || = ||\mathcal{F}(f_1) - \mathcal{F}(f_2)||$.
>     - This distance-preserving capability is beneficial in designing loss functions in a multimodal setting.
> - Invertible: The Fourier transform is not an idempotent function but periodic with a period of 4, i.e., $\mathcal{F}^4(f) = f$. This would prevent the iterative architecture outlined in Section 3 from working since a repeat application of the transform would lead to different representations. Meanwhile, the Fourier Inversion Theorem [6] shows that we can invert the frequency-domain representation to its original function without the 4x repeat application, making it suitable for the chosen iterative architecture.
> - Efficient: The Fast Fourier Transform (FFT) has a time complexity of $\mathcal{O}(n log(n))$ making it scalable to very large datasets.

---

> > ### Author Response · Authors · 2024-11-21
> >
> > **[Q4] On scalability**
> >
> > Thanks for pointing this out. All experiments run on TCGA (i.e., BLCA, BRCA, KIRP, UCEC) already contain four different modalities. While this only includes two specific data structures (tabular and imaging), the three tabular data sources are treated as separate modalities in the model. We briefly mention in L824 that all TCGA experiments use mutations (binary variables), copy number variations (categorical variables), and gene expression (continuous variables), but acknowledge that the use of Modalities={tab, img} in Table 1 is therefore ambiguous. We updated the manuscript accordingly to reflect that this is running on four modality-specific encoders. Additionally, we would like to point to the discussion in L465-478 which discusses the time complexity of scaling to a large number of modalities.
> >
> > **[Q5] On missing modality data**
> >
> > The architecture of MM-Lego is iterative. This means that, rather than using an update function applied to the Fourier-transformed representations simultaneously (as would be the case with a monolithic architecture), we update them sequentially, which keeps the data shapes intact (see Eq1 and Figure 2). This not only allows scaling to many more models/modalities (as we discussed in Q4) but is also flexible for handling missing/non-paired data. In such scenarios, typical approaches face a data-performance trade-off: Either use all the available data and impute the missing values (which introduces noise) or use a subset of the data with available data modalities, which may dramatically reduce the sample size available for training/evaluation. In contrast, MM-Lego's iterative setup allows to skip the updates from the missing data at train and inference time. Specifically, during training, this is handled by training the LegoBlocks separately before combining them into a merged/fused performant model. During inference, we skip missing modalities as a result of the iterative attention architecture. We provide evidence of the performance of MM-Lego under these circumstances in Appendix F and discuss these capabilities in detail in Section 6 (L435-454).
> >
> > **[Q6] Implementation details**
> >
> > Thank you for pointing this out. We use the same modality-wise backbones across benchmarks. More specifically, for the TCGA WSI slides, we use ResNet50 fine-tuned on Kather100k [1]. For the tabular modalities, we employ self-normalising networks [2]. We extended the discussion in the manuscript to explicitly include these details, as they are currently only available in the code repository which can be found in the supplementary materials and will be published (with further documentation) upon publication.

---

> > > ### Author Response · Authors · 2024-11-21
> > >
> > > **References**
> > >
> > > [1] Stoica, G., et al., 2024. Zipit! merging models from different tasks without training. ICLR 2024.
> > >
> > > [2] Ilharco, G., et al., 2023. Editing models with task arithmetic. ICLR 2023.
> > >
> > > [3] Yu, Z., et al., 2017. Multi-modal factorized bilinear pooling with co-attention learning for visual question answering. In *Proceedings of the IEEE international conference on computer vision* (pp. 1821-1830).
> > >
> > > [4] Zadeh, A., et al., 2017. Tensor Fusion Network for Multimodal Sentiment Analysis. In *Proceedings of the 2017 Conference on Empirical Methods in Natural Language Processing* (pp. 1103-1114).
> > >
> > > [5] Chen, R.J., et al., 2020. Pathomic fusion: an integrated framework for fusing histopathology and genomic features for cancer diagnosis and prognosis. *IEEE Transactions on Medical Imaging*, *41*(4), pp.757-770.
> > >
> > > [6] Kather, et al., 2018, 100,000 histological images of human colorectal cancer and healthy tissue.
> > > [7] Klambauer, G., et al., 2017. Self-normalizing neural networks. *Advances in neural information processing systems*, *30*.

---

> > > > ### Comment · Reviewer_fqpF · 2024-11-25
> > > > **Thanks authors' response**
> > > >
> > > > I thank the authors' response to my previous comments. The response addressed some of my concerns, while some important concerns remain that hindered me from increasing my score.
> > > >
> > > > 1.  I agree and understand the importance of combining two different network architectures for multimodal learning, and the authors claim that the previous approaches assume architecture equivalent, which is a limitation or research gap. However, if we adopt the late-fusion strategy (feature fusion), there is no need to require the network architecture to be the same. Moreover, if we directly extract the feature for each modality and then train a classifier to combine these features, there is also no need to train the framework end-to-end. Actually, the author's solution is also late-fusion. From this case, the insight or contribution of this paper is quite incremental.
> > > >
> > > > 2. I appreciate the authors' further explanation of the motivation for using frequency-domain representations. However, besides the first point "signal preserving", the other points are the **property** of Fourier transform, not the benefit or motivation of this solution.
> > > >
> > > > 3. I cannot agree that different tabular data can be regarded as different modalities. Modalities should represent different data formats or data sources, not different information. If we adopt the same criteria, histopathology can also be regarded as a combination of different modalities, as it can only have different aspects of features.
> > > >
> > > > 4. I may not fully understand your point on missing data, but it seems that the key point is that we can update the single modality encoder with the missing data pair. However, this paradigm is helpful for learning useful single-modality feature embedding but it may not be helpful in learning cross-modal interaction, which is also important in multimodal learning.

---

> ### Author Response · Authors · 2024-11-25
>
> Thank you for the prompt response and clarification of your concerns. We further address the outlined concerns (C1-C4) below.
>
> **[C1] Clarification on paper contributions**
>
> We would like to further clarify several aspects of our work and our contributions. As we discuss in our paper (L67-90, 118-161) and outlined in our previous response (Q1), there is a difference between multimodal learning (and fusion strategies) and model merging. The former refers to a general task of learning from different data modalities that capture *different information*, often from different sources and formalised differently (eg. image, table, text etc.). Here, we can broadly differentiate between early, intermediate and late fusion, which determines how and where in model architecture the *cross-modal* signal is learnt and combined. Model merging, on the other hand, traditionally focussed on improving predictive performance by combining trained models trained on the same input data (often on different tasks [1,2]) *without retraining or fine-tuning them*. This is done by linear interpolation or combination of the model’s weights (not just the outputs of the model) and is feasible in a multi-task setting, as the input data is the same across tasks, meaning that the same architecture with a different task-specific head can be used. Meanwhile, in multimodal settings, we are dealing with different input distributions and shapes for each modality, meaning that the same architecture can often not be used across modalities.
>
> Your observation "However, if we adopt the late-fusion strategy (feature fusion), there is no need to require the network architecture to be the same. Moreover, if we directly extract the feature for each modality and then train a classifier to combine these features, there is also no need to train the framework end-to-end." is partially true. Indeed, you can train a task-specific head on the combined feature representation of unimodal encoders where the original encoder weights can be both trainable or frozen. This is reflected in our experiments (see SNN+ABMIL in Table 1) and the discussion (Table 2). However, there are several reasons why the motivations of the paper are still justified:
>
> - Strictly speaking, late fusion is not a model merging method: late fusion operates on the output space (i.e., combining the outputs of the model) while model merging operates on the parameter space across the models.
> - Cross-modal interactions: a common problem with late fusion methods is that they struggle to pick up cross-modal interactions [3, 4, 5]. In some cases, this goes against the motivation of integrating multiple modalities in the first place, as we try to gain additional predictive performance from these interactions. Many intermediate fusion methods address this problem and, in fact, LegoFuse is an intermediate fusion approach.
> - Performance: Our comparison to an ensemble (similar to what the reviewer is proposing) shows that MM-Lego is more performant (see Figure 4) and robust (see Appendix F). We believe this approach to be novel and very practically relevant.
>
> **[C2] Motivations of our methods**
>
> All of the aspects highlighted in L201-222 in the paper and our earlier response (Q3) are properties of the Fourier Transform which motivate and justify our decision to work with frequency-domain representations. Specifically, besides signal preservation, the following properties are desirable for model merging:
>
> - Efficiency: It has a time complexity of $\mathcal{O}(n log(n))$ making it scalable to very large datasets, as it’s common in multimodal scenarios.
> - Invertibility: This allows us to base the design of MM-Lego on an iterative architecture, which offers additional benefits (see C4, Q2 and Section 6). It follows from the Fourier Inversion Theorem [6] that we can efficiently invert frequency-domain representation to its original function.

---

> > ### Author Response · Authors · 2024-11-25
> >
> > **[C3] Different modalities**
> >
> > In most biomedical domains, modalities are heterogeneous in how they are represented, distributed and structured, as well as the amount of information they carry [5]. As such, different modalities can represent data from different sources and can be in different formalisms - but they definitely **represent different information.** Crucially, capturing salient biological signals from different data assays (e.g., RNAseq, whole genome sequencing, microscopy for the TCGA tasks) motivates the integration of different modalities in the first place. We acknowledge that this varies from many multimodal vision & language setups, where modalities are often a different representation of the same concept (e.g., an image and text description of a scene), in L59-61 in the manuscript.
> >
> > In the context of multimodal learning, we are interested in interconnected heterogeneous modalities (interacting and/or connected via some statistical/semantic relationship). In our experiments, we use data from different sources (see below) and in different structures - but all carry different information, leading to different interactions.
> >
> > For instance, in the case of the TCGA experiments, we leverage 4 modalities: 1 imaging modality (Whole Slide Images) and 3 modalities formalised as tabular datasets: gene expression, Copy Number Variations (CNVs) and mutations. Gene expressions are measured in the RNA, while CNVs and mutations denote structural variations and small/large-scale somatic/germline changes in the DNA. All of these encompass different information from *different sources* which has been differently measured/extracted from the patients’ genomic profiles (in the case of the tabular modalities) and tissue biopsies (in the case of the image modality), providing different aspects of the disease. As such, within MM-Lego, *we model each modality separately*.  The tabular modalities are high-dimensional vectors and use a modality-specific encoder. Therefore, each modality in the TCGA experiments obtains their own encoder such that we have:
> >
> > - “WSI” encoder
> >     - Sample Input: Raw image converted to a bag of patches, $d_{wsi} = n_{patches} \times 256 \times 256 \times 3$
> >     - Encoded input using ResNet50, pre-trained on Kather100k :  $h_{wsi} = n_{patches} \times 2048$
> > - “Mutation” encoder
> >     - Non-variable mutation genes were filtered (i.e., every sample contains mutation or none contains mutation)
> >     - Input: Raw input mutation data vector for cross-attention (Figure 2)
> >     - Encoder: none as the vector is already relatively small after ETL: $d_{mut} = h_{mut} = 21$
> > - “Copy Number Variations” encoder:
> >     - Non-variable copy number genes were filtered
> >     - Input: Copy number variations for each gene (categorical variable ranging -2 (deep deletion) to +2 (strong oncogenic amplification): $d_{cnv} = 1333$
> >     - Encoder: SNN (1333 → 512): $h_{cnv} = 512$
> > - “Gene expressions” encoder
> >     - Low variable genes were filtered
> >     - Input: log1p transformed bulk RNAseq gene expression: $d_{rna} = 1558$
> >     - Encoder: SNN(1558 → 512): $h_{rna} = 512$
> >
> > Note that these relate to the Breast Invasive Carcinoma (TCGA-BRCA) patients as an example. The remaining implementation details for all TCGA and MIMIC encoders will be available in the appendix and code.
> >
> > **[C4] Missing data**
> >
> > Imagine a scenario where we have access to rich multimodal data from many patients in order to *train* a multimodal model. In healthcare, data acquisition of some modalities can be costly and/or require large manual labour. Therefore, while we can have access to such data for training a multimodal model, it is likely that there will be data from some clinics without all modalities available. This will be an issue when applying the multimodal model in practice (i.e., for inference). As we outlined in our previous response, most approaches in this scenario will face a data-performance trade-off: either use all the available data and impute the missing values (which introduces noise) or use a subset of the data with available data modalities, which may dramatically reduce the sample size available for training/evaluation. Our approach, MM-Lego, circumvents these issues via the iterative architecture and the update skips. While the resulting models can skip an update, this is far from learning from single-modality as it is applied on some of the samples (i.e. only samples w/ missing modalities)  during training/inference. Moreover, as we discuss in the paper and our previous response (Q2), MM-Lego has two variants: LegoMerge and LegoFuse. The latter, LegoFuse, is capable of learning cross-modal interactions, as they are indeed important for multimodal learning (please refer to Table 2 of the manuscript for an overview of the capabilities of MM-Lego).

---

> > > ### Author Response · Authors · 2024-11-25
> > >
> > > **References**
> > >
> > > [1] Stoica, G., et al., 2024. Zipit! merging models from different tasks without training. ICLR 2024.
> > >
> > > [2] Ilharco, G., et al., 2023. Editing models with task arithmetic. ICLR 2023.
> > >
> > > [3] Liu, Z. and Shen, Y., 2018, July. Efficient Low-rank Multimodal Fusion with Modality-Specific Factors. In *Proceedings of the 56th Annual Meeting of the Association for Computational Linguistics (Long Papers)*.
> > >
> > > [4] Nagrani, A., et al., 2021. Attention bottlenecks for multimodal fusion. *Advances in neural information processing systems*, *34*, pp.14200-14213.
> > >
> > > [5] Liang, P.P., et al., 2022. Foundations and Trends in Multimodal Machine Learning: Principles, Challenges, and Open Questions. *arXiv preprint arXiv:2209.03430*.
> > >
> > > [6] Folland, G.B., 2020. *Introduction to partial differential equations*, p.16 - Fourier Inversion Theorem. Princeton university press.

---

> > > > ### Comment · Reviewer_fqpF · 2024-11-29
> > > > **Further comments**
> > > >
> > > > Although I appreciate the author's response to my comments, I cannot agree with the response here. Here are some brief points.
> > > >
> > > > 1. Multimodal learning and model merging: "This is done by linear interpolation or combination of the model’s weights (not just the outputs of the model) and is feasible in a multi-task setting," If you really combine the weights of different models/encoders, this can be regarded as model merging not multimodal learning. However, what you have done in this paper is still "advanced feature fusion," not "advanced weight fusion." This is the reason I think you should compare multimodal learning approaches and explain your motivation from the multimodal learning aspect.
> > > >
> > > > 2. Again, "Efficiency" and "Invertibility" are some properties of the Fourier transform. It ensures this operation is efficient and can preserve the original information. But my question is, what is the additional benefit of this compared with the original feature space fusion? For example, direct feature fusion does not need Fourier transform, which is more efficient and does not need any additional computation of transform (although this point is a little tricky....)
> > > >
> > > > 3. Sorry, I cannot agree that "copy number" and "mutation status" can be regarded as different modalities from my view.
> > > >
> > > > 4. Again, I understand your approach can use single modality data or partial modality data to learn better feature extraction. But your main point in this paper is "multimodal learning." What I am not convinced about is how your approach (missing the modality part) can benefit "cross-modal interaction," not the feature encoder.

---

> > > > > ### Author Response · Authors · 2024-11-29
> > > > >
> > > > > Thank you for further clarifying your concerns. We understand your argument and would like to clarify some points further and resolve the remaining ambiguities.
> > > > >
> > > > > **[C1.1]** **Clarification on “feature fusion” vs “weight fusion”**
> > > > >
> > > > > In our paper, we propose two variants of MM-Lego - *LegoMerge (L248-264)* and *LegoFuse* (L265-275). The merging procedure in *LegoMerge* broadly consists of two steps:
> > > > >
> > > > > 1. We calculate the Harmonic mean of the latent representations $\psi(\mathcal{L})$ (Equation 4) —> we agree that this is “feature fusion”. For $m$ modalities, the harmonic mean is calculated as:
> > > > >
> > > > >     $\psi(L^{(A)}, L^{(B)}, \dots, L^{(m)}) = \left(\frac{m \prod_{i=1}^{m} |L^{(i)}|}{\sum_{j=1}^{m} \prod_{i \neq j} |L^{(i)}|}\right) \cdot e^{i \cdot \frac{\sum_{i=1}^{m} \angle L^{(i)}}{m}}$
> > > > >
> > > > > 2. We apply spherical linear interpolation on the weights of the task-specific heads. This is now possible as the latent bottleneck constraint of the *LegoBlocks* ensures equal dimensions between the latent and task-specific heads across modalities. —> this is the “weight fusion” part that qualifies MM-Lego as a model merging method. The $m$-modality case for the equation in L263 can be written as:
> > > > >
> > > > >     $w_{\text{merged}} = \frac{\sum_{i=1}^m w_i \cdot \sin\left(\theta_i \cdot \mu_i\right)}{\sin\left(\sum_{i=1}^m \theta_i \cdot \mu_i\right)}$ for interpolation coefficients $\mu_1, \mu_2, \cdots, \mu_m$ and $\sum_{i=1}^m \mu_i = 1$. Here, $\theta_i$ is the angle between each $w_i$ and a reference vector which is chosen as the average between the weight vectors.
> > > > >
> > > > >
> > > > > However, we acknowledge that step 2 in this procedure is only mentioned in L259-264 and should be made more prominent in Figure 1, which may have led to this ambiguity.
> > > > >
> > > > > **[C1.2] Comparison to multimodal learning approaches**
> > > > >
> > > > > We would like to point out that all of the multimodal baselines referenced in the paper (SNN+ABMIL (concat), SNN+ABMIL (bilinear fusion), Perceiver (early fusion), MultiModN, Multimodal Co-Attention Transformer (MCAT) and Hybrid Early Fusion (HEALNet)) present very recent multimodal learning approaches. These baselines either apply an aggregate function or gating mechanism over the modality-specific encoders (”feature fusion”) or they fit a fusion network over a combined representation of each modality. Given that *LegoFuse* also effectively fine-tunes a fusion network over the combined modality representation (derived using the harmonic mean), we believe that these are very relevant benchmarks.

---

> > > > > > ### Author Response · Authors · 2024-11-29
> > > > > >
> > > > > > **[C2] Additional benefit of the Fourier Transform**
> > > > > >
> > > > > > We agree that “invertibility” and “efficiency” are properties of the Fourier transform and we do not claim this as an innovation in the manuscript. However, they are important properties which make the Fourier transform an appropriate choice for the design of the *LegoBlocks*, which is the argument we make in our method Section 3. To clarify further:
> > > > > >
> > > > > > - **Why do we need the Fourier transform in the first place (instead of using “feature fusion”)?**
> > > > > >
> > > > > >     Several related works from different domains and modalities [1, 2, 3] emphasise the importance of fusion or aggregation methods that avoid signal interference, where meaningful patterns or complementary information from individual modalities may be attenuated, distorted, or lost due to naive aggregation approaches such as averaging or summation (”feature fusion”). We illustrate an example of signal interference of feature fusion in Appendix E which shows an example of the detrimental effect that signal interference can have when we use feature fusion. If we use average pooling (i.e., we take the mean between the two feature representations), the signal of the two modalities may cancel itself out, leading to a muted representation (Appendix E, Figure 5, bottom left). Meanwhile, if we first apply the Fourier transform to both latent representations and take the harmonic mean, this signal interference does not occur. We take advantage of this property in the first step of *LegoMerge* (as mentioned in response C1.1)*.*
> > > > > >
> > > > > >     - As such, the main motivation for using the Fourier transform is to take advantage of the orthogonal property of the Fourier transform, as mentioned in Q3 of our initial response.
> > > > > > - **Why is its invertibility important?**
> > > > > >
> > > > > >     There are two key aspects why the invertibility matters:
> > > > > >
> > > > > >     - **Tracking complex numbers**: The Fourier transform $\mathcal{F}(h)$ decomposes the latent into its frequency components which are represented by complex numbers, each of which has a real $(h^{\mathcal{F}})^r$ and imaginary component $(h^{\mathcal{F}})^i$. Most machine learning frameworks only operate on real numbers, meaning that we cannot pass the imaginary component into the cross-attention layer. However, dismissing the imaginary component entirely would discard all sine-wave contributions from the signal’s frequency domain which would lead to a drastically distorted signal after the inverse transform $\mathcal{F}^{-1}$. Therefore, we keep track of the imaginary component and use it in the inverse transform after each update. This process is illustrated in Figure 2.
> > > > > >     - **Why not apply the Fourier transform repeatedly in each layer?**
> > > > > >         - The Fourier transform is not an idempotent function but periodic with a period of 4, i.e., $\mathcal{F}^4(f) = f$. This would prevent the iterative architecture outlined in Section 3 from working since a repeat application of the transform would lead to different representations. Meanwhile, the Fourier Inversion Theorem [6] shows that we can invert the frequency-domain representation to its original function without the 4x repeat application, making it suitable for the chosen iterative architecture.
> > > > > >
> > > > > >     Taking these two aspects together, the Fourier transform’s invertibility is an important requirement for the *LegoBlock*, our key architectural component, to 1) be able to reconstruct the original representation required for iteratively passing our latent bottleneck $L$ through multiple *LegoBlocks* and 2) keep track of the imaginary component to prevent a distorted reconstruction of the original representation.
> > > > > >
> > > > > > - **Why do we care about the time complexity of the Fourier transform?**
> > > > > >
> > > > > >     Since we apply the Fourier transform multiple times in this architecture, poor time complexity would be a costly addition. However, the Fast Fourier Transform has a time complexity of $\mathcal{O}(n log(n))$ which makes it relatively cheap considering that it’s only applied to the dimensions of the latent bottleneck $L^{(m)}$ and the each modality’s embeddings $h^m$.

---

> > > > > > > ### Author Response · Authors · 2024-11-29
> > > > > > >
> > > > > > > **[C3] Treating Copy Number and Mutations as separate modalities**
> > > > > > >
> > > > > > > The term “modality” (referring to data modalities) is indeed overloaded and has several different definitions.
> > > > > > >
> > > > > > > - **How do we define a modality?**
> > > > > > >
> > > > > > >     This work follows the, widely adopted, definition from [4] which differentiates data modalities if they are heterogeneous in some of the following dimensions: element representation, distribution, structure, information, noise, and task relevance. Several other works [5, 6] use related taxonomy of what comprises a “modality”. Therefore, we believe that the terminology of copy numbers, gene expressions, and mutations as separate modalities is generally acceptable by the community in these applications.
> > > > > > >
> > > > > > >     We would like to outline the differences on some of the criteria below.
> > > > > > >
> > > > > > >     | **Criteria** | **Description** | **Mutation** | **Copy Numbers** | **Gene Expression** |
> > > > > > >     | --- | --- | --- | --- | --- |
> > > > > > >     | Element representation | Most basic unit of data that cannot be broken down into further units | Boolean | Ordinal variable (integer) | Continuous variable (float) |
> > > > > > >     | Distribution | Frequency/likelihood of elements in modalities | Sparse binary distribution, often long-tailed at the population level | Discrete multimodal distribution | After log1p transform, the gene expression values approximate a gaussian distribution per gene |
> > > > > > >     | Information | Total information content present in each modality | Low variation | Higher variation | Highest variation |
> > > > > > > - **Why is the information and task relevance between CNV and mutations heterogeneous?**
> > > > > > >
> > > > > > >     **Task relevance:** While we acknowledge that mutations and copy number variations both reflect genomic alterations, they fundamentally represent different information regarding the function and relevance to the tumour [7]. Concretely, in cancer genomics, single nucleotide variants (SNVs i.e., ”mutations”) are treated as distinct modalities from CNVs due to differences in scale, mechanisms, and impact. SNVs represent small-scale changes, often arising from replication errors or DNA repair failures and lead to precise effects such as frameshifts (e.g., BRCA1 truncations in breast cancer, which are relevant to the TCGA-BRCA task). CNVs, on the other hand, are large-scale structural alterations, such as amplifications and deletions, affecting kilobases to megabases of DNA. Functionally, SNVs often act as precise cancer drivers, whereas CNVs contribute to tumour progression through dosage effects, network disruptions, or by exposing recessive mutations. This distinction is essential for our task of survival prediction, as well as targeted therapeutic strategies, as point mutations may guide specific treatments, while CNVs inform broader genomic vulnerabilities like synthetic lethality (e.g., PARP inhibitors for BRCA1/2 loss).
> > > > > > >
> > > > > > >     **Information:** Combining these modalities can enhance tumour risk profiling by identifying synergistic alterations that jointly drive aggressive phenotypes, enabling more accurate prognostic models as exemplified by our study.

---

> > > > > > > > ### Author Response · Authors · 2024-11-29
> > > > > > > >
> > > > > > > > **[C4] Cross-modal interactions and missing data**
> > > > > > > >
> > > > > > > > **Missing data.** To further explain situations in which MM-Lego is beneficial in handling missing data, we need to distinguish between data availability at train-time vs. inference-time:
> > > > > > > >
> > > > > > > > - Training:
> > > > > > > >     - MM-Lego: The fact that the unimodal encoders can initially be trained in isolation and *later* be combined using *LegoMerge* or *LegoFuse* means that we can use unpaired data for each modality to train the encoders.
> > > > > > > >     - Others: Contrary, many of the end-to-end trained fusion models (e.g., MCAT, Late Fusion, Perceiver (early) baselines) require paired data for training of the fusion operators due to their monolithic architecture.
> > > > > > > > - Inference:
> > > > > > > >     - MM-Lego: If one modality for a sample is missing, the inference architecture for MM-Lego (see L173-186) allows the forward pass to skip missing modalities. Note that this does not mean that modalities are missing for *every* sample, but rather individual patients do not have each modality recorded, as is commonly the case in clinical practice.
> > > > > > > >     - Others: If a sample is missing a modality, we would need to instantiate the sample through e.g., through imputation to run the model’s forward pass (which can add noise). Note that the exception for this are the MultiModN and HEALNet baselines, which also use an iterative architecture.
> > > > > > > >
> > > > > > > > **Cross-modal interactions.**
> > > > > > > >
> > > > > > > > We would like to emphasise that learning cross-modal interactions is only possible with *LegoFuse.* While the merging procedure using the harmonic mean is designed to capture signals from multiple modalities, capturing their complex interactions is significantly improved by fine-tuning the model with some paired data. This happens in the fine-tuning steps of *LegoFuse* and is reflected in the improved performance of this method displayed in Table 1.
> > > > > > > >
> > > > > > > > **References**
> > > > > > > >
> > > > > > > > [1] Yu, Z., et al., 2017. Multi-modal factorized bilinear pooling with co-attention learning for visual question answering. In *Proceedings of the IEEE international conference on computer vision* (pp. 1821-1830).
> > > > > > > >
> > > > > > > > [2] Zadeh, A., et al., 2017. Tensor Fusion Network for Multimodal Sentiment Analysis. In *Proceedings of the 2017 Conference on Empirical Methods in Natural Language Processing* (pp. 1103-1114).
> > > > > > > >
> > > > > > > > [3] Chen, R.J., et al., 2020. Pathomic fusion: an integrated framework for fusing histopathology and genomic features for cancer diagnosis and prognosis. *IEEE Transactions on Medical Imaging*, *41*(4), pp.757-770.
> > > > > > > >
> > > > > > > > [4] Liang, P.P., et al., 2022. Foundations and Trends in Multimodal Machine Learning: Principles, Challenges, and Open Questions. *arXiv preprint arXiv:2209.03430*.
> > > > > > > >
> > > > > > > > [5] Baltrušaitis, T., et al., 2018. Multimodal machine learning: A survey and taxonomy. *IEEE transactions on pattern analysis and machine intelligence*, *41*(2), pp.423-443.
> > > > > > > >
> > > > > > > > [6] Lahat, D., et al., 2015. Multimodal data fusion: an overview of methods, challenges, and prospects. *Proceedings of the IEEE*, *103*(9), pp.1449-1477.
> > > > > > > >
> > > > > > > > [5] Carvalho, C.M. and Lupski, J.R., 2016. Mechanisms underlying structural variant formation in genomic disorders. *Nature Reviews Genetics*, *17*(4), pp.224-238.

---

### Official Review · Reviewer_Ph1F · 2024-11-02

**Soundness:** 3
**Presentation:** 3
**Contribution:** 4
**Rating:** 8
**Confidence:** 4

**Summary:**

This paper introduces Multimodal Lego (MM-Lego), a flexible framework for transforming any set of encoders into a multimodal model with minimal or no fine-tuning. By using a wrapper mechanism to ensure shape consistency among different modality representations, MM-Lego enables efficient fusion of representations in the frequency domain, reducing signal interference across modalities. Results on seven tasks demonstrate MM-Lego’s strong performance.

**Strengths:**

1. MM-Lego offers a versatile solution to integrate various encoders into a single multimodal model with limited fine-tuning, bringing novelty with its wrapper and frequency-domain fusion approach.

2. The paper provides an extensive comparison with existing multimodal fusion methods, addressing multiple advanced criteria and establishing MM-Lego’s relative advantages.

3. Comprehensive experiments validate MM-Lego’s performance, particularly in the biomedical domain.

**Weaknesses:**

1. A primary concern is the lack of rigorous evidence supporting the claim that frequency-domain merging avoids signal interference. Unlike data types like images or time series, frequency representations of high-dimensional embeddings lack intuitive clarity, making the necessity of this approach uncertain.

2. While the authors highlight MM-Lego’s ability to handle multiple modalities, the paper lacks experiments involving more than two modalities. This limitation leaves the model’s robustness in such scenarios untested.

**Questions:**

1. Could additional ablation studies be conducted to establish the necessity of frequency-domain representations in both LegoBlock and LegoMerge? For example, how would performance change if fusion occurred in the spatial domain?

2. Could the performance of LegoBlock be included in Table 1, not just in Table 4, to facilitate a more direct comparison in the main text? Additionally, certain phenomena, such as LegoBlock’s occasional underperformance compared to individual encoders like SNN or ABMIL, could benefit from further discussion.

3. A case study involving the fusion of more than two modalities would strengthen the paper’s claims about MM-Lego’s multimodal capabilities.

---

> ### Author Response · Authors · 2024-11-21
>
> Thank you for your thoughtful review and encouraging outlook on our work.
>
> **[Q1] On the necessity of frequency-domain representations**
>
> To emphasise the signal interference in other fusion methods in the spatial domain (compared to the MM-Lego frequency domain), we would like to further elaborate on some of the results in Table 1 (filtered below) which show the impact of the frequency-domain merge. For the four TCGA datasets, we used the SNN and ABMIL encoder wrapped in the LegoBlock and compared it to bilinear fusion (a commonly used end-to-end trained fusion method in the spatial domain). For the BLCA, BRCA, and KIRP datasets, LegoMerge (where we do not use any end-to-end training) outperforms the fusion method that requires paired training data. Meanwhile, the bilinear fusion performs worse than the best unimodal model on (BLCA, BRCA, and UCEC). These results improve further when using LegoFuse (see Table 1 of the paper).
>
> |  | **BLCA** | **BRCA** | **KIRP** | **UCEC** |
> | --- | --- | --- | --- | --- |
> | **SNN (tab)** | 0.689±0.012  | 0.544±0.020  | 0.798±0.035  | 0.589±0.057  |
> | **ABMIL (image)** | 0.591±0.057 | 0.610±0.093 | 0.741±0.080  | 0.558±0.040 |
> | **SNN + ABMIL (Bilinear)** | 0.622±0.054  | 0.557±0.089  | 0.811±0.108  | 0.666±0.031  |
> | **LegoMerge(SNN+ABMIL)** | **0.701±0.021**  | **0.601±0.025**  | **0.825±0.114**  | **0.625±0.080** |
>
> We would like to offer additional theoretical intuition on why the frequency domain representations do not suffer signal interference. Several related works in different domains and modalities [1, 2, 3] emphasise the importance of fusion or aggregation methods that avoid signal interference, where meaningful patterns or complementary information from individual modalities may be attenuated, distorted, or lost due to naive aggregation approaches such as averaging or summation. For example, bilinear pooling of two latent vectors $h_1 \in \mathbb{R}^n$ , $h_2 \in \mathbb{R}^m$ and resulting tensor $h_{12} \in \mathbb{R}^d$ ($y=h_1^T A h_2 + b$) with a learnable parameter $A \in \mathbb{R}^{n \times m \times d}$ follows the same motivation.  With the same motivation in mind, we take advantage of the Fourier transform’s orthogonal properties, as after the transform each frequency component (represented by sine and cosine waves) is orthogonal to the others, such that $\int_{-\infty}^{\infty} sin(\omega_1h_1) cos(\omega_2h_1) dh = 0$ for angular frequencies $\omega_1 \neq \omega_2$ [4]. This means that the contribution of each frequency is independent of others and there is no overlap between them. It follows that if we take the harmonic mean of two fourier-transformed latents $\mathcal{F}_1(\omega)$ and $\mathcal{F}_2(\omega)$ as $H(w) = \frac{2\mathcal{F}_1(\omega)\mathcal{F}_2(\omega)}{\mathcal{F}_1(\omega)+\mathcal{F}_2(\omega)}$, the harmonic mean aggregates each $\omega$ in a localised manner. This means that each frequency component in $\mathcal{F}_1(\omega)$ interacts only with the corresponding frequency component in $\mathcal{F}_2(\omega)$, i.e., without interference from other frequencies.
>
> Beyond these results and the theoretical grounding, which we have emphasised in the paper revision, we provide two empirical examples of signal interference in latent variables in Appendix E of the manuscript.
>
> **[Q2] Extension of Table1**
>
> Thanks for the suggestion. We updated Table1.
>
> **[Q3]** **Merging more than two modalities**
>
> Thanks for pointing this out. All experiments  performed on TCGA (i.e., BLCA, BRCA, KIRP, UCEC) already contain four different modalities. While this only contains two different heterogeneous data structures (tabular and imaging), the three tabular data sources are treated as separate modalities in the model. We briefly mention in L824 that all TCGA experiments use mutations (binary variables), copy number variations (categorical variables), and gene expression (continuous variables). Nevertheless, we acknowledge that the use of Modalities={tab, img} in Table 1 is ambiguous. We updated the manuscript accordingly to reflect that this is running on four modality-specific encoders.
>
> **References**
>
> [1] Yu, Z., et al., 2017. Multi-modal factorized bilinear pooling with co-attention learning for visual question answering. In *Proceedings of the IEEE international conference on computer vision* (pp. 1821-1830).
>
> [2] Zadeh, A., et al., 2017. Tensor Fusion Network for Multimodal Sentiment Analysis. In *Proceedings of the 2017 Conference on Empirical Methods in Natural Language Processing* (pp. 1103-1114).
>
> [3] Chen, R.J., et al., 2020. Pathomic fusion: an integrated framework for fusing histopathology and genomic features for cancer diagnosis and prognosis. *IEEE Transactions on Medical Imaging*, *41*(4), pp.757-770.

---

### Official Review · Reviewer_Qibw · 2024-11-03

**Soundness:** 3
**Presentation:** 3
**Contribution:** 3
**Rating:** 6
**Confidence:** 4

**Summary:**

The presented work introduces multimodal Lego, a novel strategy to fuse latent embeddings from different domains, such as representations learned from images, tabular or time series. The authors hypothesize that processing and fusing embeddings in the frequency domain allows them to be agnostic towards the topology of the representation space and avoid signal interference when merging embeddings. To this end, the authors propose a so-called "LegoBlock" that converts multimodal embeddings via a Fourier transform before iteratively updating them via an attention mechanism. Multiple of these modules can be combined by calculating the harmonic mean between the embeddings ("LegoMerge") or by training a cascade of modules that sequentially update the latent embedding based on information from the various domains ("LegoFuse"). The utility of the method is demonstrated on three medical datasets that each combine two data modalities. LegoMerge is shown to outperform unimodal methods, while LegoFuse outperforms various recent multimodal baselines.

**Strengths:**

- The paper researches a very timely topic, the processing and fusion of multimodal medical data. The authors compellingly motivate this in a well written introduction and background section.

- The proposed method of adapting and fusing multimodal embeddings in the frequency domain code appears to be interesting, novel, and have some theoretical grounding.

- The conducted experiments include three public datasets spanning various data domains, as well as several relevant baseline methods.

**Weaknesses:**

- The authors only provide limited theoretical motivation for their design choice of converting and processing embeddings in frequency space.

- Some parts of the proposed method are not clearly described. In particular, several aspects with regard to the architecture, functioning, training and tuning of the LegoBlocks are ambiguous or not described at all.

- Similarly, the description of the experimental setup is missing key information so that it is probably not possible to reimplement the benchmarking experiments.

- Several of the core claims made by the authors are not backed up by experimental evidence. In particular the method's ability to handle more than two modalities and its computational efficiency are not adequately evaluated.

- Finally, a meaningful ablation study is currently missing.

(All of the these perceived weaknesses are described in much greater detail in form of specific questions and suggestions in the next section.)

**Questions:**

Questions and suggestions regarding the work's motivation:

- The authors motivate processing the embeddings in frequency space by citing the advantageous properties of Fourier space with regard to its compactness, robustness against noise and robustness against signal interference. However, several of the referenced works are rather unspecific with regard to these claims and the authors do not provide detailed proof. I suggest that the authors expand the corresponding section and include more detailed arguments for this design choice.

- I believe that the statement that "LegoMerge does not require any finetuning" is slightly misleading. If I understand the method correctly the LegoBlocks are a wrapper around already existing embeddings that have to be trained themselves in order to tune (at least) the weights of the attention mechanism, $W^q_m$, $W^k_m$, and $W^v_m$. Compared to simply merging the embeddings, this does require additional fine-tuning steps. Accordingly, I do not believe that the statement that LegoMerge does not require any training time (cf. Table 3) is accurate. Consequently, I believe the paper should be adjusted to clearly reflect this.


Questions and suggestions regarding the proposed methodology:

- The paper is ambiguous on whether the raw input embeddings $h^{(A)}$ or their Fourier transform $\mathcal{F}(h^{(A)})$ are used in the attention mechanism of the LegoBlocks and the calculation of the harmonic mean during LegoMerge. Equations 3 and 4 indicate the former, while Figure 1 depicts the latter case. Please clarify this and clean up the notation.

- Also, in the right-most panel of Figure 1, why is one of the inputs to the LegoBlocks $L^{(A)}$?

- In the LegoBlocks, how is $L_0$ initialized, and is it updated during tuning?

- How is the number of update steps $S$ tuned?

- One of the main claims of the work is its robustness against different topologies of embeddings. What happens if the initial data embeddings, $X^{(A)}$, $X^{(B)}$, $X^{(C)}$, have different dimensionalities and consequently require Fourier transforms of different dimensionalities? It is not apparent to me, why embeddings of time series and tabular information should be two-dimensional as suggested by the paper.

- Why do the authors apply an inverse Fourier transform at the end of each LegoBlock only to convert it back into frequency domain for a subsequent block or the final task head? Do any artifacts occur because of this repeated conversion?


Questions and suggestions regarding the conducted experiments:

- Even after reading Appendix C and D, many aspects about the conducted experiments are unclear to me and I do not believe that it is possible for a reader to reimplement them based on the current description. For example, how were the 80,000 $\times$ 80,000 histopathology images of the TCIA dataset encoded? Which mortality was assessed during the experiments with the MIMIC-III dataset? Which model was used to encode the time series data? The authors should aim to provide much more detail with regard to the conducted experiments.

- In light of the considerable effort required to fine-tune each LegoBlock (see earlier comments), I am missing a comparable effort to tune a neural network that processes the concatenated multimodal embeddings. Analogously to the proposed method, one could employ a multi-layer perceptron with S layers and an attention mechanism at each layer. If this approach was adequately tuned, it would function as meaningful, simple and potentially powerful baseline, while simultaenously directly ablating the design choice of converting embeddings to Fourier space.

- The authors should include experiments in which three or more different modalities are included. Currently, none of the conducted experiments supports their main claim that their proposed method scales effectively beyond two modalities.

- While the authors have conducted basic experiments demonstrating their method's ability to handle missing data, the conducted experiments are rather simplistic and only included in the supplementary material. Considering they pertain to a main claim of the work, the authors should expand these experiments and work them into the main paper.

- Have the authors conducted experiments whether the order of fusion during LegoFuse influces its performance?

- What is the added benefit of including both Tables 1 and 4 instead of simply merging them?

---

> ### Author Response · Authors · 2024-11-20
> **Rebuttal**
>
> Thank you for your in-depth review and valuable feedback. We are very encouraged that you describe MM-Lego as timely, well-written, and acknowledge the core idea as interesting and novel. In the following, we provide detailed answers to your comments with additional explanations of the raised issues. We will also include these clarifications and additions in the manuscript.
>
> **[W1/Q1] Motivation for frequency-domain representations:**
>
> We would like to further elaborate on the motivation to use frequency-domain representations which the paper touches upon in L203-221. We will add a more detailed discussion in the paper appendix.
>
> - Signal preserving: The general motivation for signal-preserving aggregation for multimodal machine learning has been extensively studied in the context of fusion methods. Several related works in different domains and modalities [1, 2, 3] emphasise the importance of fusion or aggregation methods that avoid signal interference, where meaningful patterns or complementary information from individual modalities may be attenuated, distorted, or lost due to naive aggregation approaches such as averaging or summation. For example, bilinear pooling of two latent vectors $h_1 \in \mathbb{R}^n$ , $h_2 \in \mathbb{R}^m$ and resulting tensor $h_{12} \in \mathbb{R}^d$ ($y=h_1^T A h_2 + b$) with a learnable parameter $A \in \mathbb{R}^{n \times m \times d}$ follows the same motivation. With this motivation in mind, we take advantage of the Fourier transform’s orthogonal properties, as after the transform each frequency component (represented by sine and cosine waves) is orthogonal to the others such that $\int_{-\infty}^{\infty} sin(\omega_1h_1) cos(\omega_2h_1) dh = 0$ for angular frequencies $\omega_1 \neq \omega_2$ [4]. This means that the contribution of each frequency is independent of others and there is no overlap between them. If we take the harmonic mean of two fourier-transformed latents $\mathcal{F}_1(\omega)$ and $\mathcal{F}_2(\omega)$ as $H(w) = \frac{2\mathcal{F}_1(\omega)\mathcal{F}_2(\omega)}{\mathcal{F}_1(\omega)+\mathcal{F}_2(\omega)}$, the harmonic mean aggregates each $\omega$ in a localised manner. This means that each frequency component in $\mathcal{F}_1(\omega)$ interacts only with the corresponding frequency component in $\mathcal{F}_2(\omega)$, i.e., without interference from other frequencies.
> - Distance preserving:  Parseval’s Theorem shows that the Fourier transform is a unitary operator, meaning that the sum or integral of the square of a function is equal to the sum or square of its transform [4,5]. As such, the distances between two signals are the same between the transformed and untransformed representations.
>     - Concretely, the theorem states that the energy of a signal is preserved in both the time domain and frequency domain, where its energy is measured as the integral of the function. Formally $\int_{-\infty}^{\infty} |f(h)|^2dh = \int_{-\infty}^{\infty} |\mathcal{F}(f)(\omega)|^2 d \omega$ for latent signal $f(h)$ and its fourier transform $\mathcal{F}$.
>     - The Euclidean distance between two signals  $f_1(h)$ and $f_2(h)$ in the spatial domain is: $||f_1-f_2||  = \sqrt{\int_{-\infty}^{\infty}|f_1(h) - f_2(h)|^2 dh }$
>     - The Euclidean distance between the Fourier transforms of two signals is:
>     - $||\mathcal{F}(f_1) - \mathcal{F}(f_2)|| = \sqrt{\int_{-\infty}^{\infty} | \mathcal{F}(f_1)(\omega) - \mathcal{F}(f_2)(\omega)|^2 d\omega}$
>     - From Parseval’s Theorem, it follows that $||f_1 - f_2 || = ||\mathcal{F}(f_1) - \mathcal{F}(f_2)||$.
>     - This distance-preserving capability is beneficial in designing loss functions in a multimodal setting.
> - Invertible: The Fourier transform is not an idempotent function but periodic with a period of 4, i.e., $\mathcal{F}^4(f) = f$. This would prevent the iterative architecture outlined in Section 3 from working since a repeat application of the transform would lead to different representations. Meanwhile, the Fourier Inversion Theorem [6] shows that we can invert the frequency-domain representation to its original function without the 4x repeat application, making it suitable for the chosen iterative architecture.
> - Efficient: The Fast Fourier Transform (FFT) has a time complexity of $\mathcal{O}(n log(n))$ making it scalable to very large datasets.

---

> ### Author Response · Authors · 2024-11-20
>
> **[Q2] - “LegoMerge does not require any fine-tuning” is ambiguous**
>
> The authors would like to clarify the ambiguity that may arise from this statement. For MM-Lego’s architecture, we broadly have two sets of parameters:
>
> - The original encoder weights: a higher number of parameters (often 100M+ depending on the foundation model used), if we unfreeze and retrain these, we refer to this as “fine-tuning”.
> - The weights for the LegoBlock adapters: a lower number of parameters (typically <1-2M) which need to be fitted in a supervised way. This can be done in two ways: 1) retrospectively (for large foundation models) or 2) during training of the original model as a wrapper (which Table 3 refers to).
>
> The statement you are referring to is correct in the latter case where the model was already wrapped with the LegoBlock during training. When fitted retrospectively, this still uses significantly fewer parameters than when using fine-tuning. As an illustrative example, fine-tuning just the final three layers of a ViT-large model would tune about 22M parameters, while fitting the LegoBlock adapter is more parameter efficient with approx. 2M parameter. As such, the statement and values reported in Table 3 are correct in the second case, while they are not in the first. We will update Table 3 for both training modes and emphasise this important distinction.
>
> **[Q3] - $h^{(A)}$ or $\mathcal{F}(h^{(A)})$ in attention mechanism**
>
> The attention mechanism is applied in the frequency domain, as indicated in Figures 1 and 2. We agree that Equation 3 is inconsistent with that representation, which is now corrected to fall in line with Figure 2 such that $(L_{t+1}^\mathcal{F})^r = \text{softmax}\left(\frac{(L_t^\mathcal{F})^r W_m^q \cdot (\mathcal{F}(h^{(A)})^r W_m^k)^\top}{\sqrt{d_k}}\right) \cdot (\mathcal{F}(h^{(A)})^r W_m^v)$ where $\mathcal{F}(\cdot)^r$ denotes the real component of the Fast Fourier Transform.
>
> **[Q4] - $L(A)$ as LegoBlock input**
>
> This is further illustrated through Figure 2 and the architecture section in L173-185. As shown in Equation 1 and 3, each cross-attention layer takes in the $L_s$ and outputs the update $L_{s+1}$. The same update is reflected in Figure 1.
>
> **[Q5] - Initialisation of $L_0$**
>
> $L_0$ is randomly initialised as a `nn.Parameter()` with the specified latent dimensions and is with the corresponding modality in each cross-attention pass.
>
> **[Q6] - Number of update steps**
>
> Across all experiments, we use a depth of S=4 which we found to be stable. We initially ran a grid search across crucial hyperparameters for the architecture such as the number of update steps $S$ and the dimensions of $L$. We will update the Appendix with a table of all hyperparameters used for the experiments, which can also be found in the supplementary materials and further implementation details can be found in the codebase that would be made public upon publication.

---

> ### Author Response · Authors · 2024-11-20
>
> **[Q9] Implementation Details**
>
> The paper is accompanied by a code base (see zipped supplementary materials) which will make it easier for readers to reproduce the experiments. The current supplementary materials contain the main code module of the codebase and the full repository will be made available upon publication, including all details on hyperparameters and run instructions. We will also make sure to add further information that can currently only be found in the repository to the final manuscript. Answers to the additional questions:
>
> - Encoders for TCGA slides: ResNet50 fine-tuned on Kather100k [7], a large collection of histopathology data, mentioned in L838.
> - Mortality: The MIMIC-III mortality label corresponds to in-hospital mortality, i.e., to whether a patient died during the hospital stay. More information about the task label can be found on the [physionet website](https://physionet.org/content/mimiciii/1.4/).
> - Time series data: We use the same Perceiver as the one used in the Unimodal baseline. The exact Perceiver configurations can be found in the config files of the repository.
>
> **[Q10] Comparable effort with concatenated multimodal embedding**
>
> - *“[…] missing a comparable effort to tune a neural network that processes the concatenated multimodal embeddings”:* The `SNN+ABMIL (CC, Late)` baseline in Table 1 does exactly this, i.e., it is using an Attention-based Multiple Instance Learning model for the whole slide images and encodes all tabular modalities with a self-normalising network. The resulting embeddings are concatenated and passed into task-specific fully-connected layers. This baseline is tuned end-to-end (with the exception of the vision encoder).
> - *“[…] one could employ a multi-layer perceptron with S layers and an attention mechanism at each layer”*: This suggestion closely corresponds to the MultiModN [8] and HEALNet [9] baselines. MultiModN uses multiple iterative MLPs as linear projections of modalities into a shared, fixed-size space. HEALNet uses a similar iterative setup coupled with an attention layer for each update. The baselines were specifically chosen since they are a relevant comparison of an end-to-end trained model to our setup.
>
> **[Q11] Experiments with >2 modalities**
>
> All experiments run on TCGA (i.e., BLCA, BRCA, KIRP, UCEC) already contain four different modalities, namely WSIs, mutations (binary), copy number variations (categorical), and gene expression (continuous) (see L824). While this only contains two specific data structures (tabular and imaging), the three tabular data sources are treated as separate modalities in the model. We acknowledge that the use of *Modalities={tab, img}* in Table 1 is therefore ambiguous. We will update the manuscript accordingly to reflect that this is running on four modality-specific encoders.
>
> **[Q12] Missing Data Experiments**
>
> The missing data experiments are currently only implemented on the MIMIC dataset since this was the only paired dataset from our experiments that provides sufficient samples for this ablation study. The pathology datasets from TCGA have relatively few samples with all four modalities present, meaning that the subsample required to create the symmetric difference of overlapping modalities would be too small to train any stable model. However, the authors commit to looking for additional datasets that would allow for additional missing modality ablation studies in the main body before the camera-ready deadline.
>
> **[Q13] Modality order in *LegoFuse***
>
> During the development of *LegoFuse*, we performed preliminary analyses on the influence of the order of modalities on the downstream performance: We did not find any significant difference. Since all modalities are encoded in the latent array and contextualise one another as long as the number of steps is $s>1$, we have not found any evidence that the order of modalities matters.
>
> **[Q14] Combining Table 1 and Table 4**
>
> Table 1 and Table 4 present slightly different presentations on mostly identical data. The main difference is that the LegoMerge performance is presented differently in Table 4. In hindsight, the authors agree with the reviewer’s sentiment that they should be combined and will update the manuscript accordingly.

---

> > ### Author Response · Authors · 2024-11-20
> >
> > [1] Yu, Z., et al., 2017. Multi-modal factorized bilinear pooling with co-attention learning for visual question answering. In *Proceedings of the IEEE international conference on computer vision* (pp. 1821-1830).
> >
> > [2] Zadeh, A., et al., 2017. Tensor Fusion Network for Multimodal Sentiment Analysis. In *Proceedings of the 2017 Conference on Empirical Methods in Natural Language Processing* (pp. 1103-1114).
> >
> > [3] Chen, R.J., et al., 2020. Pathomic fusion: an integrated framework for fusing histopathology and genomic features for cancer diagnosis and prognosis. *IEEE Transactions on Medical Imaging*, *41*(4), pp.757-770.
> >
> > [4] Oppenheim, A.V., Willsky, A.S. and Nawab, S.H., 1997. *Signals & systems*. *Ch. 6 Fourier Transforms*, Pearson Educación.
> >
> > [5] Parseval, Marc-Antoine. Mémoire sur les séries et sur l’intégration complète d’une équation aux différences partielles linéaires du second ordre, à coefficients constants. *Mém. prés. par divers savants, Acad. des Sciences, Paris,(1)* 1, no. 638-648 (1806): 42.
> >
> > [6] Folland, G.B., 2020. *Introduction to partial differential equations*, p.16 - Fourier Inversion Theorem. Princeton university press.
> >
> > [7] Pocock, J., et al., 2022. TIAToolbox as an end-to-end library for advanced tissue image analytics. *Communications medicine*, *2*(1), p.120.
> >
> > [8] Swamy, V., et al., 2024. Multimodn—multimodal, multi-task, interpretable modular networks. *Advances in Neural Information Processing Systems*, *36*.
> >
> > [9] Hemker, K., et al., 2024. HEALNet: Multimodal Fusion for Heterogeneous Biomedical Data. *Advances in Neural Information Processing Systems*, *37*.

---

> > > ### Comment · Reviewer_Qibw · 2024-11-24
> > >
> > > I would like to thank the authors for their detailed response to the comments of the four reviewers. As a result, many of my original questions were answered and concerns resolved. Assuming the authors will faithfully implement the promised changes - in particular providing additional rationale for the shift to the frequency domain and clarifying any ambiguities regarding the required finetuning of the proposed method - I have increased my score.
> > >
> > > However, I remain unconvinced that the experiments are suited to demonstrate the utility of the proposed method. Specifically, I find that the claims regarding the method's scalability and it being agnostic towards vastly different encoding topologies are not fully supported by experimental evidence. While the authors have explained that their tabular data aggregate information from several domains, including laboratory tests and genotyping, this is substantially different than combining specialized embeddings, which are extracted from encoders that trained exclusively on a single specific domain. Ultimately, this issue is keeping me from increasing my score further to an acceptance rating.
> > >
> > > Additionally, I have two minor remaining issues. First, I believe I understand it after reading the authors’ answer to **Q4** and rereading the corresponding sections. However, I am still slightly confused, why both the stack and the weave strategy start by contextualizing $L^{(A)}$ with information from domain $A$? Second, I wanted to ask whether the authors could provide any insights into my seventh and eighth question that pertain to the dimensionality and stability of the Fourier transforms?

---

> ### Author Response · Authors · 2024-11-25
>
> Thank you very much for the prompt response to our comments and revising your original score. We would like to address the remaining concerns below:
>
> **[W1] Experiments to demonstrate the utility of the proposed method**
>
> Based on the comment, we would like to clarify a suspected misunderstanding: the current experimental setup does not use “tabular data […] to aggregate information from several domains”, i.e., the experiments do not take features from different sources and put them in a centralized feature table. For instance, in the case of TCGA, each tabular modality is a high-dimensional vector and uses a modality-specific encoder. Therefore, each modality in the TCGA experiments obtains its own encoder such that we have:
>
> - WSI encoder
>     - Sample Input: Raw image converted to a bag of patches, $d_{wsi} = n_{patches} \times 256 \times 256 \times 3$
>     - Encoded input using ResNet50, pre-trained on Kather100k :  $h_{wsi} = n_{patches} \times 2048$
> - Mutation encoder
>     - Non-variable mutation genes were filtered (i.e., every sample contains mutation or none contains mutation)
>     - Input: Raw input mutation data vector for cross-attention
>     - Encoder: none as the vector is already relatively small after ETL: $d_{mut} = h_{mut} = 21$
> - Copy Number encoder:
>     - Non-variable copy number genes were filtered
>     - Input: Copy number variations for each gene (categorical variable ranging -2 (deep deletion) to +2 (strong oncogenic amplification): $d_{cnv} = 1333$
>     - Encoder: SNN (1333 → 512): $h_{cnv} = 512$
> - Gene expressions
>     - Low variable genes were filtered
>     - Input: log1p transformed bulk RNAseq gene expression: $d_{rna} = 1558$
>     - Encoder: SNN(1558 → 512): $h_{rna} = 512$
>
> Note that these relate to the Breast Invasive Carcinoma (BRCA) patients as an example. The implementation details for all TCGA and MIMIC encoders will be available in the appendix and code.
>
> Therefore, *MM-LEGO indeed allows for combining specialized embeddings* from different encoders across the modalities, such as using (pre-trained) ResNets for image and SNN for tabular data, and using the raw input (no encoding).
>
> We hope that this addresses your primary remaining concern.
>
> **[Additional concern 1] Starting latent initialization as $L^{(A)}$**
>
> Thank you for the clarification. We recognize the confusion with the notation on the RHS of Figure 1. Indeed, the starting initialization in both the “stack” and “weave” cases should be $L_0$ rather than a domain-specific latent. This latent will then be updated in the iterative pass through the LegoBlocks. Thank you again for pointing this out - we updated the figure accordingly.

---

> ### Author Response · Authors · 2024-11-25
>
> **[Additional concern 2] Responses to Q7 and Q8 of the original questions**
>
> Apologies and thanks for pointing this out — it appears that these responses were not posted as part of our original response:
>
> **[Q7] - Different dimensions of embeddings**
>
> We would like to emphasise that the manuscript states a 1D encoding for tabular data in L302 and L840. We illustrate the Fourier transform in Equation 2 on 2D Tensors since both the image embedding of the whole slide images uses a bag of patches ($n_{patches} \times dim_{patch}$) and the latent bottleneck $L \in \mathbb{R}^{c \times d}$ is two-dimensional. For the one-dimensional tensors, the Fourier transform in Equation 2 is adjusted accordingly. Since each modality gets its own FusionBlock, the transform can be adjusted to the dimensions of each modality’s embedding, while the dimensions of the latent bottleneck remain consistent across modalities. This makes the approach flexible to work with different encoder topologies.
>
> **[Q8] Repeated Fourier and inverse transforms**
>
> No artefacts are occurring through the repeat Fourier conversion, as pointed out in the Fourier Inversion Theorem [6] mentioned in an earlier response. To the contrary, the repeat conversion is necessary to *prevent* the occurrence of artefacts during the training process:
>
> - Tracking complex numbers: The Fourier transform $\mathcal{F}(h)$ decomposes the latent into its frequency components which are represented by complex numbers, each of which has a real $(h^{\mathcal{F}})^r$ and imaginary component $(h^{\mathcal{F}})^i$. Most machine learning frameworks only operate on real numbers, meaning we cannot pass the imaginary component into the cross-attention layer. However, dismissing the imaginary component entirely would discard all sine-wave contributions from the signal’s frequency domain which would lead to a drastically distorted signal after the inverse transform $\mathcal{F}^{-1}$. Therefore, we keep track of the imaginary component and use it in the inverse transform after each update. This process is illustrated in Figure 2.
> - Artefacts through repeat transforms: As outlined in an earlier response, the FFT is a periodic transform and applying it repeatedly would lead to different representations for different layer passes of the model, making it practically impossible for the model to pick up any patterns.

---

### Official Review · Reviewer_rH9P · 2024-11-04

**Soundness:** 3
**Presentation:** 3
**Contribution:** 4
**Rating:** 6
**Confidence:** 3

**Summary:**

The authors study the multimodal learning problem and propose a novel method that can merge encoders from different modalities freely. This is achieved through their newly proposed Lego block, which iteratively applies a discrete FFT to modality-specific dataset X^m and modality-specific latent representation L^m. The authors further propose LegoMerge, which can merge Lego blocks from different modalities without any additional training, and LegoFuse, which merge Lego blocks by stacking all the blocks and fine-tuning.

**Strengths:**

1. The idea of the LegoBlocks is novel and interesting.
2. Both LegoMerge and LegoFuse show promising results.
3. The wrapper nature of LegoBlocks allows its application to wider range of tasks.

**Weaknesses:**

1. The experiments do not include the merge between image and textual data, which might be the most common/important scenario.
2. The authors state one of the advantage of the proposed method is the ability to merge more than 2 modalities, yet all experiments are conducted with only 2 modalities.

**Questions:**

1. I suggest to carry out the same type of experiments on CheXpert (https://arxiv.org/abs/1901.07031), where the downstream task can be disease classification. The experiments can be done only on the subset of CheXpert given the limited time during rebuttal.
2. If possible, I'd like to see the performance on datasets with >=3 modalities.

---

> ### Author Response · Authors · 2024-11-21
>
> Thank you for your thoughtful comments and suggestions for improvement. In the following, we respond to your comments with additional explanations of the raised issues. We also include these clarifications and additions in the manuscript.
>
> **[W1] No experiments with vision & language models**
>
> The clear focus of MM-Lego is to look at biomedical data domains where 1) modalities do not share semantics (i.e., an image and its description being a projection of the same concept), 2) we do not have a lot of paired data, and 3) we may have one-to-many cardinalities between the samples (e.g., one tissue sample with multiple DNA/RNA sequencing reads). These datasets are very common in biological and medical research domains and require fundamentally other learning paradigms to vision & text benchmarks, where most of the learning is happening on *paired* data using contrastive learning. We outline this distinction in L59-66, L166, and the discussion in L411-414, but made this aspect more prominent in the revised version of the manuscript.
>
> **[Q1] Experiments on CheXpert**
>
> Thank you for the suggestion to run experiments on CheXpert - we would like to point out that the dataset paper you mentioned is a multi-task vision-only benchmark containing *“224,316 chest radiographs of 65,240 patients labelled for the presence of 14 common chest radiographic observations”* [1]*.* While we are happy to run additional benchmarks, it is not obvious to us how this allows for further insight into the multimodal focus of the paper. This will also be less applicable to the multimodal fusion benchmarks, which expect multiple modalities (otherwise they are just a unimodal encoder).
>
> **[Q2/W2]** **Merging more than two modalities**
>
> All experiments run on TCGA (i.e., BLCA, BRCA, KIRP, UCEC) already contain four different modalities. While this only includes two specific data structures (tabular and imaging), the three tabular data sources are treated as separate modalities in the model. We briefly mention in L824 that all TCGA experiments use mutations (binary variables), copy number variations (categorical variables), and gene expression (continuous variables), but acknowledge that the use of Modalities={tab, img} in Table 1 is therefore ambiguous. We updated the manuscript accordingly to reflect that this is running on four modality-specific encoders. Additionally, we would like to point to the discussion in L465-478 which discusses the time complexity of scaling to a large number of modalities.
>
> **References:**
>
> [1] Irvin, J., et al., 2019, July. Chexpert: A large chest radiograph dataset with uncertainty labels and expert comparison. In *Proceedings of the AAAI conference on artificial intelligence* (Vol. 33, No. 01, pp. 590-597).

---

### Author Response · Authors · 2024-11-21

**For all reviewers**,

We thank all reviewers for their valuable suggestions and constructive feedback, which will significantly contribute to the improvement of our paper.

MM-Lego introduces a novel paradigm for both model merging (LegoMerge) and Fusion (LegoFuse) by learning representations in the frequency domain. This enables merging any unimodal encoders into a multimodal model with very minimal fine-tuning. We are glad that all reviewers have recognized the work’s originality, interesting domain, and good results as well as acknowledging the paper as **well-written, timely** (`Qibw`), **novel** (`Ph1F, Qibw, rH9P`), and **applicable to a wider range of tasks** (`rH9P, Ph1F`).

We are revising the paper with the reviewer’s suggestions and will share the updated version with tracked changes to the OpenReview platform shortly. In the meantime, we are looking forward to a productive discussion.

---

> ### Author Response · Authors · 2024-11-27
>
> **For all reviewers,**
>
> We would like to thank all reviewers who have participated in the discussion so far and send a friendly reminder to those who have not. We uploaded a revised version of the manuscript that incorporates the changes requested by all reviewers, highlighted in blue.
>
> We are looking forward to a productive discussion for the remaining period.

---

### Author Response · Authors · 2024-12-02

Dear Reviewers,

Thank you very much for the constructive discussion up to this point. We appreciate how much time it takes to appropriately engage with a paper during the review process and are grateful that you have done so.

We have tried our best to address your open questions and concerns over the past weeks. As the discussion period comes to an end, we are keen to clarify any remaining issues and **kindly remind you to update your scores if we were able to address your concerns**.

---

### Meta-Review · Area_Chair_LGyg · 2024-12-20

**Metareview:**

Key Strengths:
1. The paper presents a novel approach to multimodal data fusion using frequency-domain processing, with promising results across multiple datasets
2. The "wrapper" nature of LegoBlocks makes it adaptable to various tasks and architectures
3. The paper includes comprehensive comparisons with existing methods and evaluations on public datasets
4. The writing quality is good, particularly in the introduction and background sections

Major Concerns:
1. Limited Experimental Validation.
2. Insufficient theoretical justification for using frequency-domain processing for embedding fusion
3. Experimental setup lacks sufficient detail for reproducibility

**Additional Comments On Reviewer Discussion:**

The paper received mixed reviews and the authors were able to address some of the concerns raised by the reviewers. While there is no final consensus, the AC acknowledges both the merits outlined by the positive reviewers & shortcomings of the paper.
The authors are suggested to provide a substantial improvement in final version to clarify main issues raised by reviewers, i.e.,
- present the motivation for adopting Fourier transform for model fusion
- demonstrate the novelty or advancement of the proposed "model merge" strategy
- the claimed scalability when it comes to integrating multimodal models.

---

### Decision · Program_Chairs · 2025-01-22

Accept (Poster)